



# An updated tropospheric chemistry reanalysis and emission estimates, TCR-2, for 2005-2018

Kazuyuki Miyazaki[1,2], Kevin Bowman[1], Takashi Sekiya[2], Henk Eskes[3], Folkert Boersma[3,4], Helen Worden[5], Nathaniel Livesey[1], Vivienne H. Payne[1], Kengo Sudo[6,2], Yugo Kanaya[2], Masayuki Takigawa[2], and Koji Ogochi[2]

[1]Jet Propulsion Laboratory, California Institute of Technology, Pasadena, CA, USA
[2]Earth Surface System Research Center, Japan Agency for Marine-Earth Science and Technology (JAMSTEC), Yokohama, 236-0001, Japan
[3]Royal Netherlands Meteorological Institute, De Bilt, the Netherlands
[4]Wageningen University, Environmental Sciences Department, Wageningen, the Netherlands
[5]Atmospheric Chemistry Observations & Modeling (ACOM), National Center for Atmospheric Research (NCAR), Boulder, CO, USA
[6]Graduate School of Environmental Studies, Nagoya University, Nagoya, Japan

**Correspondence:** Kazuyuki Miyazaki(kazuyuki.miyazaki@jpl.nasa.gov)

**Abstract.** This study presents the results from the Tropospheric Chemistry Reanalysis version 2 (TCR-2) for the period 2005-2018 at 1.1° horizontal resolution obtained from the assimilation of multiple updated satellite measurements of ozone, CO, $NO_2$, $HNO_3$, and $SO_2$ from the OMI, SCIAMACHY, GOME-2, TES, MLS, and MOPITT satellite instruments. The reanalysis

calculation was conducted using a global chemical transport model MIROC-CHASER and an ensemble Kalman filter technique that optimizes both chemical concentrations of various species and emissions of several precursors, which was efficient for the correction of the entire tropospheric profile of various species and its year-to-year variations. Comparisons against independent aircraft, satellite, and ozonesonde observations demonstrate the quality of the reanalysis fields for numerous key species on regional and global scales, as well as for seasonal, yearly, and decadal scales, from the surface to the lower strato-

sphere. The multi-constituent data assimilation brought the model vertical profiles and inter-hemispheric gradient of OH closer to observational estimates, which played an important role in improving the description of the oxidation capacity of the atmosphere and thus vertical profiles of various species. The evaluation results demonstrate the capability of the reanalysis products to improve understanding of the processes controlling variations in atmospheric composition, including long-term changes in near-surface air quality and emissions. The estimated emissions can be employed for the elucidation of detailed distributions of

the anthropogenic and biomass-burning emissions of co-emitted species ($NO_x$, CO, $SO_2$) in all major regions, as well as their seasonal, and decadal variabilities. The datasets are available at: https://doi.org/10.25966/9qgv-fe81 (Miyazaki et al., 2019a).



## 1 Introduction

As a consequence of rapid global economic development, along with governmental regulations, air pollutant emissions have
been changing dramatically in many regions (e.g. Zhang et al., 2016; Mi et al., 2017; Zheng et al., 2018). These emission
changes have led to substantial variations in air quality and climate over the past decades. A long-term record of atmospheric
composition is essential to comprehend the impact of human activity and natural processes on the atmospheric environment
and its effect on air quality, human health, ecosystems, and climate. Various measurements have been employed for assessing
geographical, vertical, and temporal variations in atmospheric composition. However, the present in-situ observing network
(e.g. Schultz et al., 2017) is primarily clustered in the US, Europe, and East Asia and therefore insufficient for global air quality
assessment. Satellite measurements have immense potential for complementing in-situ measurements in providing data on the
global and regional distributions of air pollutants in the atmosphere; however, they address complex vertical sensitivities for
many key species. The evaluation of global atmospheric composition fields with a suite of satellite measurements is challenging
because of different vertical sensitivity profiles, various overpass times, and mismatches in spatial and temporal coverage
between the instruments (Boersma et al., 2016).

Among many species that degrade air quality and contribute to climate change, tropospheric ozone is one of the most
important air pollutants and greenhouse gases in the atmosphere (e.g. Stevenson et al., 2013; Myhre et al., 2013). Tropospheric
ozone also plays a crucial role in the oxidative capacity through the production of hydroxyl radicals (OH) (e.g. Logan et al.,
1981; Thompson, 1992). However, ozone is not emitted directly but formed through secondary photochemical production
from precursors, including hydrocarbons or carbon monoxide (CO), in the presence of nitrogen oxides ($NO_x$). These ozone
precursors are largely controlled by anthropogenic and natural emissions, e.g., transportation, industry, lightning, biogenic and
biomass-burning sources. Analyses of co-emitted species have been used to explain emission and ozone production processes
(e.g. Mauzerall et al., 1998; Ryerson et al., 1998).

Emission inventories have been developed to assess the impact of human and natural activities on the atmospheric environ-
ment. Bottom-up inventories have struggled to account for these changes, leading to substantial errors in emission factors and
activity rates especially in developing countries. Using satellite data, previous studies have shown increases in $NO_x$ emissions
between 2005 and 2010 and a rapid reduction after 2011 in China (Qu et al., 2017; Miyazaki et al., 2017; Zheng et al., 2018),
decreasing CO emissions from the United States and China between 2001 and 2015 (Jiang et al., 2017), a drastic $SO_2$ emission
decrease since 2007 for China (Li et al., 2017), and a slowdown in the U.S. $NO_x$ emissions in recent years (2011–2015) (Jiang
et al., 2018). An important outcome of these studies is the realization of the importance of background chemical conditions,
(i.e., ambient ozone, $NO_x$, and VOC) to accurately quantify the emissions-to-concentration relationship. Consequently, it is
critical to incorporate multiple constituents to accurately represent these conditions.

Chemical data assimilation can help mitigate the limitations of current observing systems using models to propagate obser-
vational information in time and space from a limited number of observed species to a wide range of chemical components,
including surface concentrations and emissions (e.g. Lahoz and Schneider, 2014). Reanalysis is a systematic approach to create
a long-term data record consistent with model processes and observations, using data assimilation. To improve the understand-





ing of emission variability and the processes controlling the atmospheric composition, chemical reanalysis products have been generated by integrating various satellite measurements. Using an ensemble Kalman filter (EnKF) data assimilation technique, Miyazaki et al. (2015) simultaneously estimated concentrations and emissions of various species for an eight-year tropospheric

chemistry reanalysis (TCR-1) for the years 2005-2012. The TCR-1 framework based on the AGCM-CHASER (Sudo et al., 2002) and MIROC-CHASER (Watanabe et al., 2011) models has been used to provide comprehensive information on atmospheric composition and emission variability (Miyazaki et al., 2012a, 2014, 2017; Miyazaki and Eskes, 2013; Ding et al., 2017). Employing the ECMWF′s Integrated Forecasting System (IFS), three recent reanalyses have also been released: the MACC reanalysis for the years 2003-2012 (Inness et al., 2013), the CAMS-Interim reanalysis for the years 2003-2018 (Flem-

ming et al., 2017) and recently the CAMS reanalysis for the years 2003 to present (Inness et al., 2019). A decadal reanalysis of CO was conducted at NCAR (Gaubert et al., 2016).

Miyazaki et al. (2020) developed a multi-constituent multi-model chemical data assimilation (MOMO-Chem) framework that directly accounts for model error in transport and chemistry by integrating a portfolio of forward chemical transport models into an EnKF system. The MOMO-Chem framework generates an ensemble of data assimilation analyses to provide

integrated unique information on the tropospheric chemistry system including precursor emissions and their uncertainty ranges due to model errors. In spite of substantial model forecast differences, the multi-constituent assimilation was sufficient to reduce the multi-model spread for many key species. Harnessing assimilation increments in both NOx and ozone in MOMO-Chem, Miyazaki et al. (2020) also demonstrated fundamental differences in the representation of fast chemical and dynamical processes among the models.

Recently, an updated chemical reanalysis (TCR-2) has been developed based on an improved EnKF data assimilation system (Miyazaki et al., 2019a) and evaluated against independent observations for limited time periods in the KORUS-AQ aircraft campaign during Apr-May 2016 (Miyazaki et al., 2019b; Thompson et al., 2019) and over remote oceans using ship-borne measurements for the years 2012-2017 (Kanaya et al., 2019). The TCR-2 performance for 2007 has also been extensively evaluated against various independent observations within the MOMO-Chem framework (Miyazaki et al., 2020). Huijnen

et al. (2019) quantitatively compared the TCR-2 with operational CAMS reanalyses (Flemming et al., 2017; Inness et al., 2019) but for ozone only. In this study, we present the detailed evaluation results of the TCR-2 performance for the years 2005-2018 for many chemically reactive species and aerosols in the troposphere, from the surface to the lower stratosphere, at daily to decadal scales.

## 2 Data assimilation system

This section provides the details of the TCR-2 approach for 2005-2018. Table 1 compares the configurations of TCR-1 and TCR-2 systems. The major updates in TCR-2 from TCR-1 are a change in the chemistry-transport model, increased model resolution, and updated retrievals used in the assimilation.





## 2.1 Forecast model

The forecast model, MIROC-CHASER (Watanabe et al., 2011), contains detailed photochemistry in the troposphere and strato-
sphere by simulating tracer transport, wet and dry deposition, and emissions. The model calculates the concentrations of 92
chemical species and 262 chemical reactions (58 photolytic, 183 kinetic, and 21 heterogeneous reactions). Its tropospheric
chemistry considers the fundamental chemical cycle of $O_x$-$NO_x$-$HO_x$-$CH_4$-CO along with oxidation of non-methane volatile
organic compounds (NMVOCs) to properly represent ozone chemistry in the troposphere. MIROC-CHASER has a T106 hor-
izontal resolution ($1.1°$ x $1.1°$) with 32 vertical levels from the surface to 4.4 hPa. This is coupled to the atmospheric general
circulation model MIROC-AGCM version 4 (Watanabe et al., 2011). The simulated meteorological fields were nudged toward
the six-hourly ERA-Interim (Dee et al., 2011).

The a priori surface emissions of $NO_x$ and CO were obtained from bottom-up emission inventories. Anthropogenic $NO_x$
and CO emissions were obtained from the HTAP version 2 for 2010 (Janssens-Maenhout et al., 2015), which combines re-
gional inventories of the European Monitoring and Evaluation Programme (EMEP), Environmental Protection Agency (EPA),
Greenhouse Gas-Air Pollution Interactions and Synergies (GAINS), and Regional Emission Inventory in Asia (REAS). For
biomass burning were emissions, we employed the monthly Global Fire Emissions Database (GFED) version 4 (Randerson
et al., 2015). Emissions from soils were based on monthly mean Global Emissions Inventory Activity (GEIA) (Graedel et al.,
1993). Lightning $NO_x$ sources were simulated using the convection scheme of MIROC-AGCM and the relationship between
lightning activity and cloud top height (Price and Rind, 1992).

## 2.2 Data assimilation method

Data assimilation applied here is based upon on an EnKF approach, the Local Ensemble Transform Kalman Filter (LETKF)
(Hunt et al., 2007). The EnKF uses an ensemble forecast to estimate the background error covariance matrix and generates an
analysis ensemble mean and covariance that satisfy the Kalman filter equations. In the forecast step, a background ensemble,
$\boldsymbol{x}_i^b (i = 1, ..., k)$, is obtained from the evolution of an ensemble model forecast, where $\boldsymbol{x}$ represents the model variable, $b$ is the
background state, and $k$ is the ensemble size (i.e., 32 in this study). The observation operator $H$ is applied to the background
ensemble to converted them into the observation space, $\boldsymbol{y}_i^b = H(\boldsymbol{x}_i^b)$, which is composed of a spatial interpolation operator
and a satellite retrieval operator. The satellite retrieval operator uses an a priori profile and an averaging kernel of individual
measurements (e.g. Eskes and Boersma, 2003; Jones et al., 2003). Using the covariance matrices of observation and background
error as estimated from ensemble model forecasts, the data assimilation determines the relative weights given to the observation
and the background and then transforms a background ensemble into an analysis ensemble, $\boldsymbol{x}_i^a (i = 1, ..., k)$.

In the data assimilation analysis, a covariance localization and inflation was applied. The covariance localization was used to
neglect the covariance among unrelated or weakly related variables, which results in removing the influence of spurious corre-
lations resulting from the limited ensemble size. The covariance localization was also applied to avoid the influence of remote
observations that may cause sampling errors. The covariance inflation was employed to inflate the forecast error covariance,
in order to prevent underestimation of background error covariance and filter divergence caused by sampling errors associated





with the limited ensemble size and by model errors. The state vector includes several emission sources (surface emissions of $NO_x$ and CO, and lightning $NO_x$ ($LNO_x$) sources) as well as the concentrations of 35 chemical species. The emission estimation is based on a state augmentation technique, in which the background error correlations determine the relationship between the concentrations and emissions of related species for each grid point. We employed a scheme to correct diurnal

emission variability from the simultaneous assimilation of multiple satellite measurements obtained at different overpass times (Miyazaki et al., 2017). The simultaneous assimilation of multiple-species data and the simultaneous optimization of the concentrations and emission fields are important to propagate the observational information between various species and modulate the chemical lifetimes of many species, as demonstrated in our previous studies (Miyazaki et al., 2012b, 2015, 2019b).

## 3 Observations

### 3.1 Assimilated data sets

An observation operator is applied to assimilate individual measurements to map the model fields into the retrieval space. The operator includes the spatial interpolation operator, a priori profile for the satellite retrievals, and averaging kernel. See Miyazaki et al. (2020) for more details.

### 3.1.1 OMI, GOME-2, SCIAMACHY $NO_2$

The tropospheric $NO_2$ column retrievals used were the QA4ECV version 1.1 level 2 (L2) product for OMI (Boersma et al., 2017a), GOME-2 (Boersma et al., 2017b), and SCIAMACHY (Boersma et al., 2017c). The ground pixel sizes of the OMI, GOME-2, and SCIAMACHY retrievals are 13km×24km, 80km×40km, and 60km×30km, with local equator overpass times of 13:45, 09:30, and 10:00, respectively. Since December 2009, approximately half of the pixels of the OMI measurements have been compromised by the so-called row anomaly, which were excluded before data assimilation. The GOME-2 measurements

were assimilated after January 2007, whereas the SCIAMACHY retrievals were assimilated before February 2012. Low-quality data were excluded by applying the provided quality flag. A super-observation approach was employed to generate representative data with a horizontal resolution of the forecast model for OMI, GOME-2, and SCIAMACHY observations, following the approach of Miyazaki et al. (2012a). Super-observations were generated by averaging all data located within a super-observation grid cell. The detailed error characteristics and validation results of the $NO_2$ products are described by

Boersma et al. (2018).

### 3.1.2 TES ozone

The Tropospheric Emission Spectrometer (TES) ozone retrievals used are the version 6 level 2 nadir data obtained from the global survey mode (Bowman et al., 2006; Herman and Kulawik, 2013) (https://tes.jpl.nasa.gov/data/products/level-2). This data set consists of 16 daily orbits with 5×8 km footprints spaced approximately 200 km apart along the orbit track, with

the equator crossing local times of 13:40 and 02:29. Low-quality data were excluded using the quality flag information. The



availability of TES measurements is strongly reduced after 2010, which can affect the reanalysis performance (Miyazaki et al., 2015).

### 3.1.3 MOPITT CO

The MOPITT total column CO data used were the version 7 L2 TIR/NIR product (Deeter et al., 2017). The TIR/NIR product
provides the greatest sensitivity to CO in the lower troposphere and increases sensitivity to near surface CO compared to the TIR-only product. The total column averaging kernel was used in the observation operator. The reported retrieval error was used in the observation error. The super-observation approach was also applied to MOPITT observations.

### 3.1.4 MLS ozone and $HNO_3$

The Microwave Limb Sounder (MLS) data used were the version 4.2 ozone and $HNO_3$ L2 products (Livesey et al., 2011,
2018). We used MLS data for pressures of lower than 215 hPa for ozone and 150 hPa for $HNO_3$, while excluding tropical-cloud-induced outliers. The provided accuracy and precision of the measurement error were used in the observation error.

### 3.1.5 OMI $SO_2$

The OMI $SO_2$ data used were the planetary boundary layer vertical column $SO_2$ L2 product obtained with the principal component analysis algorithm (PCA) (Krotkov et al., 2016). Only clear-sky OMI $SO_2$ data (cloud radiance fraction < 20%)
with solar zenith angles less than 70° were used, following the procedure of Fioletov et al. (2016, 2017). Because of the lack of information regarding the observation error, we assumed the OMI $SO_2$ error to be a constant value of 0.25 DU, which is about half of the standard deviation of the retrieved columns over remote regions (Li et al., 2013).

## 3.2 Validation data sets

### 3.2.1 TES PAN

We use version 7 TES PAN retrievals (Payne et al., 2014; TES Science Team, 2016; Payne et al., 2017) to evaluate tropospheric profiles of PAN for years 2005-2009. TES PAN data have provided information on the long-range transport of $NO_x$ at low temperatures and ozone production in warmer regions of the remote troposphere (Jiang et al., 2016). Low-quality data were excluded using the provided quality flag and information. Payne et al. (2014) showed that the detection limit for a single TES measurement is dependent on atmospheric and surface conditions as well as on the instrument noise. For observations where
the cloud optical depth is less than 0.5, the TES detection limit for PAN is within the region of 200 to 300 pptv.

### 3.2.2 AIRS/OMI ozone

We used the joint AIRS/OMI version 1 L2 tropospheric ozone profile product (Fu et al., 2018) for 2006-2010 and 2015-2018 to evaluate decadal changes in tropospheric ozone. The ozone profile retrievals were performed by applying the JPL MUlti-SpEctra, MUlti-SpEcies, Multi-Sensors (MUSES) algorithm to both AIRS and OMI level 1B (L1B) spectral radiances (Fu



et al., 2018). The AIRS/OMI ozone profile products have been produced with a spatial sampling and the retrieval characteristics of ozone profiles equivalent to TES L2 standard data product, demonstrating the feasibility of extending the TES L2 data record by a multiple spectral retrieval approach. The retrievals show reasonable agreement with WOUDC global ozonesonde measurements (Fu et al., 2018). The AIRS/OMI data has been successfully assimilated to improve the tropospheric ozone analysis over East Asia during the KORUS-AQ campaign (Miyazaki et al., 2019b) and could be used to improve decadal ozone

reanalyses.

### 3.2.3 WOUDC Ozonesonde data

We used ozonesonde observations taken from the World Ozone and Ultraviolet Radiation Data Center (WOUDC) database (available at http://www.woudc.org) to validate the vertical ozone profiles. All available data from the WOUDC database were used (a total of 39,959 profiles for 149 stations during 2005-2018). To compare ozonesonde measurements with the reanalysis

fields, the reanalysis and model fields were linearly interpolated to the time and location of each measurement using the two-hourly output data, with a bin of 25 hPa.

### 3.2.4 WDCGG CO data

The CO concentration observations were obtained from the World Data Centre for Greenhouse Gases (WDCGG) operated by the World Meteorological Organization (WMO) Global Atmospheric Watch programme (http://ds.data.jma.go.jp/gmd/

wdcgg/). Hourly and event observations from 59 stations for 2005-2014 were used to validate the surface CO concentrations.

### 3.2.5 HIPPO aircraft data

HIAPER Pole-to-Pole Observation (HIPPO) aircraft measurements provide global information on vertical profiles of various species over the Pacific (Wofsy et al., 2012). Latitudinal and vertical variations in ozone and CO obtained from the five HIPPO campaigns (HIPPO I-V) were used to validate the assimilated profiles.

For comparison with aircraft observations (Sections 3.2.5, 3.2.6, and 3.2.7), all observed profiles were binned on a common pressure grid with an interval of 30 hPa and mapped with a horizontal resolution of 0.5° x 0.5°. The characteristics of the aircraft measurements vary significantly among different profiles; e.g., between rural and urban and between in-cloud and clear sky observations. Case-dependent evaluations would provide deeper insights into the processes and reanalysis performance in possible future studies.

### 3.2.6 NASA aircraft campaign data

Vertical profiles of nine key gases ($O_3$, CO, $NO_2$, PAN, OH, $HO_2$, $HNO_3$, $CH_2O$, and $SO_2$) were used, obtained from the following eight aircraft campaigns.

The DC-8 measurements obtained during the Intercontinental Chemical Transport Experiment Phase B (INTEX-B) campaign over the Gulf of Mexico (Singh et al., 2009) were used for the comparison for March 2006. Data collected over highly





polluted areas (over Mexico City and Houston) were removed from the comparison, as they could cause significant errors in the representativeness (Hains et al., 2010).

The Arctic Research of the Composition of the Troposphere from Aircraft and Satellites (ARCTAS) mission (Jacob et al., 2010) was executed during two three-week deployments based in Alaska (April 2008, ARCTAS-A) and western Canada (June-July 2008, ARCTAS-B). During ARCTAS-A, most of the measurements were collected between 60°N and 90°N, whereas

during ARCTAS-B, the measurements were mainly recorded in the sub-Arctic between 50°N and 70°N.

During the Deriving Information on Surface Conditions from Column and Vertically Resolved Observations Relevant to Air Quality (DISCOVER-AQ) campaign over Baltimore in the United States during July 2011, the NASA P-3B aircraft performed extensive profiling of the optical, chemical, and microphysical properties of aerosols (Crumeyrolle et al., 2014).

The Deep Convective Clouds and Chemistry (DC3) experiment field campaign investigated the impact of deep, mid-latitude

continental convective clouds during May and June 2012 over northeastern Colorado, western Texas to central Oklahoma, and northern Alabama (Barth et al., 2015). Observations obtained from the DC-8 (DC3-DC8) and Gulfstream-V (DC3-GV) aircraft were used.

The Korea-United States Air Quality (KORUS-AQ) campaign was conducted during the period May-June 2016 over the Korean peninsula. We used DC-8 aircraft measurements from 23 flights, as in our previous study (Miyazaki et al., 2019b).

The Studies of Emissions and Atmospheric Composition, Clouds and Climate Coupling by Regional Surveys (SEAC4RS) campaign was conducted over North America during August and September 2013. The DC-8 employed in situ and remote sensing instruments for radiation, chemistry, and microphysics in the southeastern U.S. from the boundary layer to the upper troposphere. All DC-8 data were used in this study.

### 3.2.7 ATom aircraft data

ATom-1 and ATom-2 flew transects through the Pacific, Southern, Atlantic, and Arctic oceans with the NASA DC-8 aircraft in August 2016 and February 2017, respectively. The 11 flights for each campaign sampled air profiles by frequently ascending and descending between 0.2 km and 12 km. The DC-8 carried a suite of instruments that measured over 100 different chemical constituents, aerosol particle properties and chemical composition. We used the merged ATom-1 and ATom-2 OH data (Wofsy et al., 2018). The same data were used in Wolfe et al. (2019).

### 3.2.8 Surface aerosol measurements

We used the in situ surface observations of sulfate, nitrate, and ammonium aerosols from the European Monitoring and Evaluation Programme (EMEP; http://ebas.nilu.no) for Europe, the Clean Air Status and Trends Network (CASTNet; https://www.epa.gov/castnet) for the United States, and the Acid Deposition Monitoring Network in East Asia (EANET; https://www.eanet.asia) for East Asia. The observation data at 52, 51, and 30 monitoring sites were obtained from the EMEP, CAST-

Net, and EANET networks for 2005-2017, respectively.





## 4 Evaluation results

This section presents validation results of numerous species using various independent observations. To confirm improvements in the reanalysis, results from a model simulation without any chemical data assimilation (i.e., a control run) are likewise shown.

### 4.1 Ozone

#### 4.1.1 Ozonesonde

Figs 1 and 2 compare the vertical profile and time series of tropospheric ozone with the global ozonesonde observations taken from the WOUDC network. The validation of the reanalysis and control run with global ozonesonde observations is summarized in Table 2. The model bias in the lower and middle troposphere is negative except near the surface at low and mid latitudes, whereas it is positive in the upper troposphere and lower stratosphere (UTLS) for the globe. The large positive biases in the extratropical UTLS could be associated with errors in the stratosphere-troposphere-exchange (STE) processes and chemical processes such as halogen chemistry, in addition to errors in the prescribed ozone concentrations above 70 hPa in the model.

The reanalysis shows improved agreement with the ozonesonde observations over the globe. The data assimilation generally decreased the ozone concentration in the extratropics UTLS (200-90 hPa) for the globe and in the middle and upper troposphere (500-200 hPa) at high latitudes of both hemispheres throughout the year. In the lower troposphere (850-500 hPa), the data assimilation increased the ozone concentrations and removed the most model negative biases for the globe. Consequently, the reanalysis mean bias became nearly zero in the extratropical UTLS regions and less than 15% in the free troposphere for the globe. At high latitudes, the tropospheric ozone is not directly constrained by any measurements. Nevertheless, the reanalysis ozone shows improved agreements with the ozonesonde measurements through atmospheric transport from lower latitudes and from the stratosphere. In the lower troposphere, the annual mean reanalysis ozone bias is less than 1.2 ppbv, except for the tropics (4.2 ppb), which is 70-94% smaller than the bias in the control run. In the middle and upper troposphere, the mean ozone bias is less than 5.7 ppbv for the SH high latitudes and 3.1 ppbv for other regions, which is 74-99% lower than the bias in the control run. The RMSE is also reduced by 6-50% for 850-500 hPa and 500-200 hPa, with large reductions for the SH mid and high latitudes (42-50%) for 500-200 hPa, except for the tropical lower troposphere. The mean bias (RMSEs) reductions in the UTLS regions are about 93-99% (51-74%) in the extratropics and 56% (41%) in the tropics.

Both seasonal and interannual variations are well reproduced by the chemical reanalysis throughout the troposphere, with temporal correlations greater than 0.90 at the mid-latitudes of both hemispheres and greater than 0.85 at NH high latitudes. The correlations in the UTLS range from 0.88 to 0.99. The lower correlations in the tropical lower troposphere (r = 0.73-0.77) with enhanced biases in winter and at SH high latitude's lower troposphere (r = 0.75) could be attributed to the remaining model errors and the lack of direct observational constraints at high latitudes throughout the reanalysis period and in the tropical troposphere after 2009 (c.f., Section 7.1). During 2005-2009, the mean ozone bias in the tropical troposphere did not change significantly with year, which suggests that the TES measurements provide constraints on making stable long-term analysis of





the free-tropospheric ozone. The observed trend is positive at the NH mid-latitudes in the lower troposphere (+0.9 ppb/year),

corresponding to increased concentrations after 2012, but the significance of this trend is not very high. The reanalysis (+0.4 ppb/year) shows better agreement with the observed slope than the control run (-1.4 ppb/year). The long-term trends will further be discussed in Section 6.

The mean ozone biases in TCR-2 are reduced from those in TCR-1 for many regions, especially for the NH mid and high latitudes (e.g., from -3.9 to -1.2 ppb and from -8.0 to -0.2 ppb at the NH high-latitudes between 850 and 500 hPa and

between 500 and 200 hPa, respectively) and SH mid latitudes (from -1.0 to 0.4 ppb and from -1.9 to -0.2 ppb between 850 and 500 hPa and between 500 and 200 hPa, respectively). An exception is the tropics, where the reduced number of ozonesonde observations for the most recent years used in the TCR-2 validation affected the evaluated performance. Huijnen et al. (2019) and Christophe et al. (2019) compared tropospheric ozone reanalysis products from CAMS, CAMS-Interim, TCR-1 and TCR-2. The updated reanalyses (CAMS-Rean and TCR-2) showed substantially improved agreements with independent ground

and ozonesonde observations over their predecessor versions (CAMS-iRean and TCR-1) for the diurnal, synoptical, seasonal, and decadal variability. The improved performance can be attributed to a mixture of various upgrades, such as revisions in the chemical data assimilation, including the assimilated measurements and the forecast model performance. The updated chemical reanalyses agree well with each other in most cases, which highlights the usefulness of the current chemical reanalyses in a variety of studies.

### 4.1.2 AIRS/OMI satellite retrievals

Fig. 3 compares the time series of ozone with the AIRS/OMI retrievals over selected polluted areas between 700 and 500 hPa during 2005-2018. The AIRS/OMI data was not available for some part of the time period (2005 and from 2011 to 2014) at the time of this study. To provide continuous decadal records of the control run and reanalysis fields in the AIRS/OMI observation space, we applied the 2007 AIRS/OMI retrieval sampling and averaging kernel to the control run and reanalysis

fields for 2005 and 2011-2014. In the United States and China, the control run generally underestimates compared with the AIRS/OMI observations especially in summer, and the reanalysis shows improved agreements. The mean bias and RMSEs over China are reduced by 80% (to -1.1 ppb) and 63% (to 4.3 ppb), respectively. Over India, the data assimilation reduced the mean bias from -9.9 ppb in the control run to 0.6 ppb, while showing larger concentrations (by about 3 ppb) after 2015 than before 2009 similar to the observations (by about 6 ppb). For tropical regions, the overall model negative biases compared to

the AIRS/OMI observations are greatly reduced in the reanalysis (e.g., from -9.9 ppb in the control run to 0.6 ppb over central Africa). The estimated reanalysis errors are mostly within the AIRS/OMI retrieval uncertainty. These improved agreements in the reanalysis, along with the good agreements between the reanalysis and ozonesonde observations (c.f., Section 4.1.1), demonstrate the great potential of AIRS/OMI data to further improve decadal ozone reanalysis, as will be discussed in Section 7.3.





### 4.1.3 Aircraft

The reanalysis captured the observed latitudinal-vertical distributions by the HIPPO aircraft measurements over the Pacific. (Fig. S1 and Table S1). On average, the control run shows negative biases in the lower troposphere (850-500 hPa) from the SH high latitudes to NH high latitudes (-4.6 to -3.6 ppb), whereas the model bias is positive in the middle and upper troposphere (19.6 to 42.6 ppb between 500-200 hPa) except in the tropics (-2.6 ppb). The negative model biases in the lower troposphere are greatly reduced by data assimilation (by 33-80%). Data assimilation introduced a slight positive bias in the NH lower troposphere, probably associated with corrections made to precursors' emissions over East Asia and the stratospheric concentrations. The positive model biases in the middle and upper troposphere (500-200 hPa) are also reduced by 44-92% except for the tropics and SH low latitudes, as commonly suggested by comparisons to the ozonesonde measurements (c.f., Section 4.1.2). These results demonstrate that the assimilation of multiple-species data sets is a powerful way to globally constrain the entire tropospheric ozone profile, including that over remote oceans.

The comparison with the NASA aircraft data (Fig. 4) shows that the control run generally underestimates ozone in the free troposphere, with largest biases (up to about 15 ppb) for the DC3-GV over the United States and KORUS-AQ profiles over South Korea. In turn, it is overestimated in the lower stratosphere for the ARCTAS-A and -B profiles over the Arctic. Near the surface, the control run overestimates ozone for the DISCOVER-AQ profile by 15 ppb and for the KORUS-AQ profile by 7 ppb, which could partly be attributed to the model representative error. Data assimilation mostly removed the model biases throughout the troposphere and lower stratosphere, even for the profiles without the direct tropospheric ozone constraints by the TES measurements (after 2009). Miyazaki et al. (2019b) demonstrated that strong corrections for the entire tropospheric ozone profile during the KORUS-AQ were mainly obtained from the combined assimilation of UTLS $O_3$ (MLS) and tropospheric $NO_2$ column (OMI and GOME-2) retrievals.

## 4.2 NO$_2$

### 4.2.1 Satellite retrievals

Fig. 5 compares the global maps of tropospheric $NO_2$ columns between the satellite measurements, control run and chemical reanalysis. The control run generally underestimated tropospheric $NO_2$ columns over most polluted areas, with large negative biases over industrial areas (e.g. East China, Europe, eastern USA, and South Africa) and over large biomass-burning areas (e.g. Central Africa). As an exception, positive model biases appeared over parts of China, mainly over southeastern China, after 2015 (Fig. 6) associated with the use of the 2010-year HTAP v2 inventories and because emission reductions after 2012 are not described by the inventories. Compared to the control run in TCR-1, the control run in TCR-2 shows reduced annual mean model biases in tropospheric $NO_2$ columns against the satellite measurements for the same time periods by up to 90% over China, by 13% over the western United States, and by 37% over South Africa, mainly attributed to the increased horizontal resolution (Sekiya et al., 2018). The different model bias against the three retrievals can be attributed to the overpass time difference and diurnal variations in chemical processes and emissions, as the three products are generated using the same retrieval approach (Boersma et al., 2018).



The negative model bias over these regions is greatly reduced in the reanalysis, decreasing the global mean negative bias by about 84-93% as compared to the three satellite retrievals to -0.03-0.02 $\times 10^{15}$ molec cm$^{-2}$ (Table 3). Data assimilation im-

provement is also observed in the reduced global RMSE from 0.30-0.38 to 0.17-0.27 $\times 10^{15}$ molec cm$^{-2}$ and in the increased spatial correlation from 0.95-0.96 to 0.97-0.98. The remaining errors in the reanalysis are considerably smaller for most polluted regions in TCR-2 than in TCR-1 (the global mean biases are -0.18 to -0.05 $\times 10^{15}$ molec cm$^{-2}$, the RMSEs are 0.38-0.95 $\times 10^{15}$ molec cm$^{-2}$, the spatial correlations are 0.92-0.97 in TCR-1). The improvements from TCR-1 can be associated with various reasons: increased model resolution, improved assimilated retrievals including reduced uncertainty for polluted regions, and

improved data assimilation setting including the use of the diurnal emission variability correction scheme.

Fig. 6 shows the time series of regional mean tropospheric NO$_2$ concentrations. The regional error statistics compared to the OMI retrievals are summarized in Table 4. Over East China, the model negative bias is relatively large in winter, particularly in comparison with SCIAMACHY and OMI during 2010-2014 when the observed NO$_2$ concentrations are relatively high. In contrast, the model bias against OMI and GOME-2 is negative during 2015-2018, when the observed concentrations are

relatively low. The reanalysis captures the observed decadal changes (r = 0.99 for OMI using montly mean concentrations), through corrections made to NO$_x$ emissions. Slight negative biases remain during the 2010-2014 winters compared with OMI and SCIAMACHY.

Over Europe, the negative model bias is persistent against the three retrievals throughout the reanalysis period. The data assimilation reduced about 30-60% of the model negative bias compared with OMI (by 54 % for mean) and most of the biases

against SCIAMACHY. In contrast, the reanalysis reveals excessively high NO$_2$ compared with GOME-2 during summer. The observed negative trend by OMI (-1.2%/year) is efficiently captured by the reanalysis (-1.2%/year, r = 0.95).

Over the United States, the observed NO$_2$ concentrations decreased rapidly during 2005-2009 and subsequently show weaker reductions, as discussed by Jiang et al. (2018). The observed negative trends (-2.3%/year) during 2005-2018 are better represented by the reanalysis (-2.1%/year) than by the control run (0.6%/year). The model negative biases compared with the

OMI measurements remain partially in late-winter and spring. Data assimilation also increased the temporal correlation with OMI from 0.54 to 0.88.

Over India, the model negative bias increased with year because of the lack of the emission increases in the a priori emissions. The reanalysis shows positive trends over the 14 years (+1.3%/year) consistent with the OMI observations (+1.6%/year), with high temporal correlations with respect to all the retrievals (r = 0.96 for OMI). The mean bias was reduced by about 80%

compared to OMI.

Over north and central Africa, and south America, the control run largely underestimated the NO$_2$ concentrations in the biomass burning off-seasons, while the interannual variability in the active seasons for biomass burning are not efficiently captured. The data assimilation removed most of the model negative bias throughout the year, except for reduced concentrations over South America in 2005, 2007, and 2010. The reanalysis period mean biases against OMI are reduced by more than 90%

over North Africa, Central Africa, and over South America, with increased temporal correlations from 0.92-0.97 to 0.98-0.99. Data assimilation reduced the seasonal amplitude by about 20-30 % over north and central Africa.





Over Southeast Asia and Australia, the control run underestimated the $NO_2$ concentrations throughout the year with respect to all the retrievals. The data assimilation removed most of the negative biases and reproduced inter-annual variability such as high concentrations in 2010 and 2013-2016 over Southeast Asia and in 2006-2007 and 2012-2013 over Australia.

Over Southern Africa, the control run underestimated the $NO_2$ concentrations by a factor of about two throughout the year, while about 50% of the model negative bias is removed by data assimilation. The remaining model errors can be partially attributed to the limitations in assimilated measurements (e.g., coverage and uncertainty) and persistent model errors, such as too-short lifetime of $NO_x$ through processes such as $NO_2 + OH$ reactions and the reactive uptake of $HO_2$ and $N_2O_5$ by aerosols (e.g. Lin et al., 2012; Stavrakou et al., 2013). Further, any errors in the location of individual sources such as power
plants in the bottom-up inventories could prevent data assimilation improvements in our approach.

### 4.2.2   Aircraft

Compared with the vertical $NO_2$ profiles from the aircraft measurements, the simulated $NO_2$ concentration in the free troposphere is generally too low, whereas the model biases within the boundary layer vary among campaigns (Fig. 4). The relatively coarse resolution of the model could cause large differences near the surface, especially at urban sites. For the ARCTAS pro-
files, the control run failed to reproduce the enhanced concentrations in the boundary layer, and data assimilation only has small impacts throughout the troposphere. In the lower stratosphere, the MLS $O_3$ and $HNO_3$ data assimilation effectively corrects the amount of $NO_2$ for the ARCTAS-A profile. The insufficient corrections at high latitudes in the troposphere are associated with limited influences of surface $NO_x$ emissions on the $NO_2$ profiles. For the DISCOVER-AQ profile, the control run overestimated rapid $NO_2$ increases toward the surface, whereas the reanalysis shows improved agreements. Compared with
the two DC3 profiles, both the control run and reanalysis show close agreement with observations from the surface to middle troposphere, while underestimating the $NO_2$ concentrations in the upper troposphere. For SEAC4RS, the data assimilation leads to an underestimation within the boundary later. In contrast, for KORUS-AQ, the negative model bias (up to about 40 %) in the boundary layer is mostly removed by data assimilation.

### 4.3   CO

### 4.3.1   Surface

We used the WDCGC in-situ measurements in 59 stations to evaluate the reanalysis CO concentrations. The comparison results are summarized in Table 5 and shown in Fig. 7 for selected sites. The control run underestimated the mean CO concentrations by 9.4 and 19.8 ppbv at NH mid and high latitudes, with the largest negative biases in winter. The model CO underestimations in the NH are commonly reported in many models (e.g. Stein et al., 2014). The model bias is positive in the tropics and SH by
about 13-14 ppbv. After data assimilation, the model biases are greatly reduced in the SH, the tropics, and NH mid-latitudes (by 66-88%), while reproducing the observed seasonal and inter-annual variations for many sites. In contrast, at NH high latitudes, the reanalysis CO in TCR-2 reveals small corrections. For instance, over Barrow, Heimaery, and Cold Bay, most of the negative





model biases remain. This is different from the substantial improvements found for the entire globe in TCR-1 (Miyazaki et al., 2015). There are several potential reasons for the remaining negative biases, as will be discussed in Section 7.4.

### 400 4.3.2 Aircraft

Both the control run and reanalysis captured latitudinal variations in CO over the Pacific acquired by HIPPO observations, including maximum gradients around the equator and the subtropical jet (Fig. S2). As summarized in Table S2, the control run underestimates CO concentrations in the NH and overestimates them in the tropics and SH almost for the entire troposphere over the Pacific. The assimilation decreased CO concentrations and removed the model positive bias by about 63-79% in the
lower troposphere and by 56-67% in the middle and upper troposphere in the tropics and SH. In the NH, data assimilation improvements are small, which can be attributed to remaining errors in the surface emissions, chemical productions and losses (i.e., OH), long-range transport from the Eurasian continent, and stratosphere-troposphere exchange (STE) (c.f., Section 7.4).

The control run generally captured the observed profiles for most NASA aircraft flights, except for an up to 50-130 ppb underestimation in the lower and middle troposphere for the ARCTAS-A, ARCTAS-B, and KORUS-AQ profiles (Fig. 4).
Substantial reductions in the model negative bias are found for the KORUS-AQ profile because of increased local and remote (mainly China) CO emissions (Miyazaki et al., 2019b). In contrast, the bias reduction is small for the ARCTAS profiles. MOPITT data are assimilated equatorward of 65°, which limits improvements at high latitudes. Meanwhile, the along-track measurements could not be representative of the concentrations within the large domain of the western Arctic during ARCTAS-B (Bian et al., 2013), which may also explain the large negative bias for the ARCTAS-B profile.

### 415 4.4 SO$_2$

Compared with the aircraft measurements (Fig. 4), the control run mostly overestimates SO$_2$ concentrations in the lower troposphere by a factor of 2-5 for the DC-3, SEAC4RS, and KORUS-AQ profiles. The data assimilation greatly reduced the positive model biases and reproduced the observed profiles, with mean bias reductions of up to 90%, mainly because of reduced surface SO$_2$ emissions. The near surface SO$_2$ concentrations became too low by about 20% after data assimilation for the
SEAC4RS and KORUS-AQ profiles, which could be associated with the large uncertainty and assumptions made (e.g., constant observation errors, c.f., Sec. 3.1.5) in the assimilated OMI SO$_2$ retrievals and possible overestimation of an atmospheric sink of SO$_2$ within the boundary layer in the model. Any errors in the assimilated retrievals and model processes could introduce biases in the estimated emissions (c.f., Section 5.3). For the ARCTAS profiles, both the control run and reanalysis show excessively low SO$_2$ concentrations throughout the troposphere, likely associated with the lack of observational constraints
and large uncertainty in the model processes.





### 4.5 PAN

#### 4.5.1 Aircraft

Comparisons with the aircraft measurements (Fig. 4) revealed that the control run tends to underestimate PAN in the free troposphere during INTEX-B by about 50-70%, during ARCTAS-B by about 10-30%, during SEAC4RC and KORUS-AQ by

up to about 30%, and for the DC3-DC8 profile by up to about 10%. Data assimilation mostly removed the negative model biases in the free troposphere. In the lower troposphere, the reanalysis reveals improved agreements for the KORUS-AQ, SEAC4RS, and DISCOVER-AQ profiles. These improvements demonstrate that information obtained from the $NO_2$ retrievals was propagated efficiently to the NOy budget for many regions. In contrast, for the INTEX-B, ARCTAS-A, and ARCTAS-B profiles, the reanalysis shows poor agreement likely due to model errors in the boundary layer chemical production and loss

processes of PAN as well as spatial representativeness errors.

#### 4.5.2 Satellite retrievals

The TES PAN retrievals allow evaluation of the conversion process from $NO_x$ to PAN and its long-range transport across both polluted and remote regions. Because of the TES single-footprint detection limit of 200-300 pptv (Payne et al., 2014), we focus on over and downstream of highly polluted areas only. Fig. 8 shows the seasonal variations of tropospheric PAN

averaged over the years 2005-2009 between 800-400 hPa. The observed PAN concentrations are the largest in boreal spring (MAM) over most polluted regions of the NH extratropics, including north America, and northern and eastern parts of the Eurasian continent, while the enhanced concentrations over northern Pacific suggests long-range transports. The springtime maximum over the Arctic can be attributed to the transport of pollution and fires from Russia and China (Fischer et al., 2014). During summer, the strong contrast between source and remote areas could reflect the short lifetime of PAN due to thermal

decomposition. The observed PAN concentrations in the tropics are high over northern Africa in DJF and over central Africa in JJA, corresponding to the biomass burning season. The enhanced concentrations over the Atlantic in SON are likely associated with lightning $NO_x$ sources, as well as strong biomass burning emissions over the Amazon and long-range transport along westerly jets.

  The control run captured well the observed spatial and temporal variability, including the enhanced concentrations over

polluted areas with a maximum in spring in the subtropics and extratropics of both hemispheres and the signals of inter-continental transports across the northern Pacific and Atlantic. The overall good agreement demonstrates the capability of the model in representing the global nitrogen cycles, as shown in the GEOS-Chem simulations (Jiang et al., 2016). Despite the good agreements, the control run is lower than the TES retrievals over eastern China and North America in SON and DJF, South America in MAM and JJA, the Middle East in JJA, and is higher over Europe and over East Asia in JJA.

Data assimilation generally increased PAN over and downstream of major polluted areas throughout the year, corresponding to the increased surface and lightning $NO_x$ emissions. The increases are large over northern and central Africa, South America and the tropical Atlantic in SON, Southeast Asia in MAM, and at the NH mid latitudes over land in MAM and JJA. The increased concentrations reduced the model negative bias against the TES retrievals over East Asia and North America in SON

 

and over Southeast Asia in DJF. These corrections led an about 4% global RMSE reduction in DJF and MAM, while the spatial pattern is reasonably captured by the reanalysis (r = 0.52-0.84). In contrast, the data assimilation adjustment increased the positive model bias in JJA over most of the NH mid latitudes and over northern and central Africa throughout the year. The remaining discrepancies can be attributed to unconstrained model processes, such as overestimated conversions from $NO_x$ to PAN and underestimated thermal decompositions, as well as the TES retrieval errors. Fischer et al. (2014) suggested that PAN is generally more sensitive to NMVOC emissions than $NO_x$ emissions. Fu et al. (2008) and Fischer et al. (2014) also suggested that underestimations in Asian outflow can be attributed to emissions of aromatic species and missing NMVOC emissions in China. Thus, adding constraints in the reanalysis framework, especially on VOCs emissions, would benefit improving PAN and chemically-related species including ozone. Further investigation on the detailed PAN distributions using aircraft and satellite measurements would be helpful to comprehend the possible mechanisms and error sources in the reanalysis PAN fields.

## 4.6 OH

OH is directly linked to the concentrations of species determining the primary production ($O_3$ and $H_2O$), removal (CO and methane), and regeneration of OH ($NO_x$). Because of the multi-constituent constraints for many key species, a positive impact is expected on global OH fields, given that the reactions are reasonably well described by the model. As shown in Fig. 9, the global tropospheric OH distribution is substantially modified in the reanalysis. Data assimilation mostly increased OH, with the largest increases in the SH tropics. The mean OH concentration in the SH tropics is increased over the reanalysis period by 20-25% at 700 hPa and 30-45% at 500 hPa. In the NH extratropics, the OH increases are about 15-20% at 700 hPa and 20-30% at 500 hPa. These increases are found throughout the reanalysis period, with the largest increases during spring-summer in both hemispheres. Both the concentration assimilation and the emission optimization were important in introducing these OH changes. The 14-year mean NH/SH OH ratio in the chemical reanalysis is 1.19±0.015 (1σ inter-annual variability), in contrast to 1.30 in the control run, which is closer to the estimates of 0.97±0.12 based on methyl chloroform observations (Patra et al., 2014). The NH/SH ratio is maximum in 2016 (1.23), reflecting relatively high OH concentrations over East Asia and low concentrations over South America (Figs. S3 and S4). The inter-annual variability can be associated with both human and natural activities, through changes in climate condition including lightning and in anthropogenic emissions, as discussed in Murray et al. (2013) and Rowlinson et al. (2019).

The tropospheric mean OH concentrations averaged during are estimated at 8.7 $10^5$molec cm$^{-2}$ for the control run and 11.5 $10^5$molec cm$^{-3}$ for the reanalysis. By applying the obtained tropospheric OH burden to the ACCMIP multi-model mean estimates of tropospheric chemical methane lifetime ($\tau_{OH}(chemical)$) from Voulgarakis et al. (2013) for 2000 (mean OH concentrations =11.7±1.0×$10^5$molec cm$^{-3}$, $\tau_{OH}(chemical)$=9.3±1.6 yr, and total life time ($\tau_{OH}(total)$)= 8.6 yr)), we estimated $\tau_{OH}(chemical)$ for 2005-2018 at 12.5 yr for the control run and 9.5 yr for for the reanalysis. The large changes in methane lifetime has a strong implication into the methane budget estimate including emission inversions.

The model bias against the aircraft profiles varies largely among the campaigns. For the INTEX-B profile, the control run captured the observed profile well, whereas the data assimilation puts too high OH throughout the troposphere, likely corresponding to increased ozone. For the ARCTAS-A, ARCTAS-B, and KORUS-AQ profiles, the model negative bias is strongly





reduced by data assimilation in the free troposphere, mainly due to the increased $NO_x$ emissions and resultant increased ozone. The large negative bias near the surface remains for the ARCTAS-B profiles. Remaining large errors in $HO_2$ could influence the performance of the simulation of OH for some profiles, including ARCTAS-B. Observed OH concentrations are also largely uncertain (e.g. Heard and Pilling, 2003; Stone et al., 2012). Brune and Thames (2019) estimated the absolute accuracy for aircraft HOx measurements to be $\pm$ 32 % at 2 sigma confidence.

The ATom measurements provide great data to evaluate remote tropospheric OH, for instance, that derived from OMI $CH_2O$ measurements (Wolfe et al., 2019). Compared with all the profiles during the ATom-1 and ATom-2, the RMSE is reduced by up to 30% above about 600 hPa in the reanalysis than in the control run (Fig. 10a). Improved agreements can be found for many profiles throughout the troposphere (Fig. S5), whereas a few profiles (e.g., on August 6, 2018, February 2, 2017, and February 5, 2017) led to a degradation in the agreements in the lower troposphere. The ATom measurements provide comprehensive pictures of inter-hemispheric ratios of OH and its seasonal changes over remote oceans (Fig. 10b,c). The observed inter-hemispheric ratio is about two near the surface and exceeds seven in the middle troposphere in boreal summer (ATom-1), whereas it is 0.4-0.8 throughout the troposphere in boreal winter (ATom-2). The control run mostly overestimated the ratios by a factor of up to 2.5 for ATom-1 and by up to 1.6 for ATom-2, with the largest overestimation in the lower troposphere. Data assimilation decreased the ratio and shows improved agreements from the surface to the upper troposphere. Because the chemical lifetimes of many species are affected by the amount of OH, the improved representation of OH profiles and its global distributions suggests that multi-constituent assimilation improves the simulation of concentrations and emissions of various species. Decadal changes in the tropospheric OH derived from the reanalysis will be discussed in Section 6.

## 4.7 Aerosols

Although no aerosol observations were assimilated, improved representations of aerosol fields can be expected through corrections made to trace gas concentrations, such as $NO_x$ and $SO_2$, that affect the formation of secondary aerosols. Figure 11 shows the scatter plots of ammonium ($NH_4$), nitrate ($NO_3$), and sulfate ($SO_4$) aerosols from in-situ observations, control run, and reanalysis. The control run overestimates ammonium and sulfate aerosol concentrations and underestimates the nitrate aerosol concentrations for most of the CASTNET (the US), EANET (East Asia), and EMEP (Europe) sites, while the estimated mean biases (Table 6) are dominated by large biases for a few stations. The median biases are lower than the mean biases for many cases. The multi-constituent data assimilation substantially modified the aerosol concentrations. The RMSE is decreased by 7-61% for ammonium aerosols, 2-11% for nitrate aerosols, and by 5-38% for sulfate aerosols, by data assimilation while the correlation improved for many cases, for instance, from 0.27 to 0.42 for ammonium aerosols compared to the EMEP observations. The median bias also became smaller (by up to 75 %) for most cases. For urban stations, the model representativeness errors may prevent data assimilation improvements, which may have caused degradation for some cases. An assessment of global particulate nitrate and ammonium aerosols in the MIROC-CHASER simulation is also given in (Bian et al., 2017).

Substantial changes in the aerosol concentrations suggest considerable potential of trace gas data assimilation for constraining secondary aerosol formation processes. Among numerous factors, optimizations of $NO_x$ and $SO_2$ emissions are considered to be essential to improve secondary aerosol formation in our framework. Our comparisons show improved agreements against





various aircraft measurements for many key species relevant to aerosol formations, such as $NO_2$, $HNO_3$ and $SO_2$ (c.f., Section 4.8). Meanwhile, assimilation of AOD and aerosol observations are required to further improve the representation of primary aerosol emissions and concentrations (e.g. Yumimoto et al., 2017). Simultaneous assimilation of trace gas and aerosol obser-

vations would be a powerful approach to fully represent aerosol-gas interactions in the data assimilation framework, which would improve both trace gas and aerosol data assimilation analysis.

## 4.8  Other reactive species

As shown in Fig. 4, the observed main structures of $HNO_3$ are generally captured well by the control run, with increasing errors toward the surface for some profiles. The increase in $HNO_3$ toward the surface is driven mainly by the oxidation of $NO_x$

in polluted areas for most profiles except for the ARCTAS-A. The control run overestimated the lower tropospheric $HNO_3$ concentrations by a factor of more than 2 for the INTEX-B, DISCOVER-AQ, and KORUS-AQ profiles, whereas it mostly underestimated $HNO_3$ concentrations throughout the troposphere for the ARCTAS-B, DC3-DC8, DC3-GV, and SEAC4RS profiles. For ARCTAS-A, the control run largely overestimated $HNO_3$ above about 600 hPa. The data assimilation mostly increases the $HNO_3$ concentrations throughout the troposphere except for the ARCTAS-A and DISCOVER-AQ profiles, pri-

marily attributing to the increased $NO_x$ emissions and $NO_2$ concentrations. The data assimilation increase largely reduced the negative model biases in the free troposphere and lower stratosphere for the ARCTAS-A and DC3-DC8 and DC3-GV profiles. In contrast, it increased the positive model bias in the lower troposphere for INTEX-B and KORUS-AQ. For DISCOVER-AQ, the positive model bias in the lower troposphere is greatly reduced. To improve the lower tropospheric $HNO_3$ concentrations, corrections for its removal processes including deposition and direct assimilation of tropospheric $HNO_3$ measurements could

be important. In the UTLS region, the MLS $HNO_3$ data assimilation mostly removed the positive model bias in $HNO_3$ for the ARCTAS-AQ profile.

The tropospheric $HO_2$ profiles mainly reflect variations in water vapor. The control run generally overestimates $HO_2$ throughout the troposphere except for the ARCTAS-B profile. The control run overestimates the tropospheric $HO_2$ concentrations. As an exception, the model negative bias is found in the lower troposphere for the ARCTAS-B and in the upper

troposphere for KORUS-AQ. Data assimilation slightly increases $HO_2$ in the lower troposphere and decreases it in the upper troposphere for most cases. The increased $HO_2$ in the lower troposphere could be associated with increased CO through the reaction of OH with CO that converts OH into $HO_2$. The remaining errors could be associated with model errors for instance in the heterogeneous loss of $HO_2$ on cloud droplet.

Both the control run and reanalysis capture well the tropospheric $CH_2O$ profiles for most campaigns, except for the

ARCTAS-A where $CH_2O$ concentrations are underestimated by a factor of 2 throughout the troposphere. The data assimilation influences on $CH_2O$ concentrations are small, because of the lack of assimilation of direct measurements. Inter-species correlations with $CH_2O$ in the in the state vector was neglected for all the assimilated measurements, which also prevented data assimilation adjustments to $CH_2O$.



## 5  Emission sources

In this section, we briefly describe the estimated emissions from the TCR-2 calculations. Further detailed analyses of the 14-year variations in the estimated emission sources and its influences on global air quality and health impacts will be discussed in a separate study. The global distribution of a priori and a posteriori emissions and its time series are shown in Figs 12 and 13 and summarized in Table 7. The estimated linear trends are shown in Fig. 14. The regional total emission statistics for surface emissions and lightning $NO_x$ sources are summarized in Table 8 and 9, respectively.

### 5.1  Surface $NO_x$ emissions

The multi-constituent data assimilation framework has been used to improve estimates of global $NO_x$ emissions (Miyazaki and Eskes, 2013; Miyazaki et al., 2014, 2015, 2017, 2019b). In this framework, the simultaneous optimization of concentrations and emissions of many species reduces the model-observation mismatches that arise from model errors other than those related to emissions. Meanwhile, the simultaneous assimilation of multiple satellite measurements obtained at different overpass times

was employed to constrain diurnal emission variability (Miyazaki et al., 2017). Thus, the estimated emissions in TCR-2 can be expected to provide unique information on decadal changes in anthropogenic and natural emission sources.

The global surface $NO_x$ emissions averaged over the 14 years are estimated at 49.2 TgN/yr from the data assimilation, which is 17.4% larger than the a priori emissions (41.9 TgN). The mean total emissions are estimated at 29.0 TgN (12.0% larger than the a priori emissions) for the NH (20°N-90°N), 16.8 TgN (22.6% larger) for the tropics (20°S-20°N), and 3.5 TgN (29.6%

larger) for the SH (20°S-90°S), as summarized in Table 7.

Data assimilation largely increased surface $NO_x$ emissions over major polluted areas such as most parts of China, Southeast Asia, and Europe (Fig. 12). The increments vary from year-to-year over these regions. For instance, they decreased in more recent years over China. This is associated with the assumptions applied to the a priori emissions, such as the use of 2010 anthropogenic emissions in the estimations for 2011-2018. The complex spatial structure of the increments over India and eastern

China suggests that the emissions evolved differently among the grid points while the bottom-up inventories exhibited large uncertainty. The seasonal variations are also largely modified for many regions. The bottom-up inventories did not consider seasonal variability for anthropogenic emissions, such as emissions from wintertime heating. Over agriculture and desert areas such as the western and central United States, Sahara, western China, and southern Europe, the summertime large positive increments can be attributed to underestimated soil emissions, as commonly suggested by Oikawa et al. (2015) and Visser

et al. (2019). By applying the ratio of different emission categories within the a priori emissions for each grid point, the global total a posteriori $NO_x$ emissions by soils is estimated at 8.7 TgN, which is about 58% larger than the a priori emissions (5.5 TgN) and closer to other recent estimates of around 10 TgN (Steinkamp and Lawrence, 2011; Hudman et al., 2012; Vinken et al., 2014). The large positive increments over north and central Africa, South America, and Southeast Asia suggest general underestimations in biomass burning emissions in the GFED v4 inventories.

Fig. 15 depicts the decadal trend of the estimated $NO_x$ emissions over major polluted regions, updated from our previous estimates (Miyazaki et al., 2017). The detailed spatial maps of the $NO_x$ emissions for an individual year are shown in Fig.



S6, and the regional yearly emission values are summarized in Table S3. For China, the estimated emissions increased from 2005 to 2011 by 30 % and decreased rapidly after 2013. Since 2016, the Chinese country-total emissions started to increase again, while exhibiting substantial spatial differences in the estimated trends. For India, the emissions show a continuous increase by 30% over 14 years. The Middle East also exhibits an emissions increase from 2005 to 2014 of 24%. After 2014, it exhibits a flattened or a slight negative trend, however with substantial spatial variations. For the United States, the emissions show a reduction of 25% from 2005 to 2010. The emission reduction is slowed down afterwards, as suggested by Jiang et al. (2018) using our previous emission estimates based on the TCR-1 system (Miyazaki et al., 2017). The estimated emissions for Europe show a negative trend during 2005-2014 (by 13%), followed by a flattened trend. Our estimates also reveal substantial emission increases for most parts of Southeast and South Asia, and Mexico after 2014. In spite of the substantial changes for many regions reflecting a combination of effects of environmental policies and economic activities, the global total emissions did not change obviously over 2005-2018 (49.2±2.8 TgN).

## 5.2 Surface CO emissions

The 14-year mean of global total emissions of CO is increased by 26% by data assimilation (1104 Tg CO vs. 877 Tg CO), which is attributable to a 35% increase in the NH and 18% increase in the tropics. The large positive increments are found over eastern and southern China, northern parts of Southeast Asia, India, and central Africa (Fig. 12). The emissions increase in the NH is large in the boreal late winter-spring period, especially over polluted areas at NH mid-latitudes, which enhanced the seasonal amplitude for industrialized countries.

The estimated emissions show strong negative trends over most parts of China (by -0.6%/yr), Japan (-2.2%/yr), Europe (-0.8%/yr) and the United States (-1.8%/yr) and positive trends over India (1.5%/yr) during 2005-2018 (Fig. 14). As seen in the underestimated decreasing trends of surface CO concentration for the NH in the current estimates (c.f., Section 4.3), the obtained CO emissions could underestimate a long-term decreasing trend in CO emissions in NH as compared with other estimates (e.g. Jiang et al., 2017). For biomass burning areas, such as Southeast Asia, Amazon, and central and north Africa, the estimated emissions exhibit a strong year-to-year variability, such as enhanced emissions over Southeast Asia in 2006-2007 and 2015, and over South America in 2007 and 2010 (Fig. S7). The regional total surface CO emission values are summarized in Table S4.

In the multi-constituent data assimilation framework, the assimilation of non-CO observations influences various chemical species including OH, which provides additional constraints on the CO emission estimation. As suggested in Section 4.6, possible underestimations in OH in the control run could lead to underestimations in the estimated CO emissions for many regions. Assimilation of ozone and $NO_2$ measurements exerts a substantial influence on OH and thus on CO emission estimates. Nevertheless, insufficient corrections for the NH extratropical CO suggest requirements for further improving CO emission estimates, as will be discussed in Section 7.4.





## 5.3 Surface SO$_2$ emissions

The 14-year mean global total surface SO$_2$ emissions are decreased by about 30% by data assimilation from 50.9 to 35.1
TgS, with large reductions in the NH (by 37%). The negative increments are also large over China (by -50%), India (-64%),
and Southeast Asia (-75%), suggesting overestimated emissions in the bottom-up inventories for most industrialized areas. In
contrast, the mean increments are positive over western Europe and the western United States (by up to 50%). These large
adjustments suggest large uncertainties in the current inventories, as suggested by Koukouli et al. (2018) and Miyazaki et al.
(2019b). The increments changed largely during the 14 years for many regions, according to substantially temporal changes in
the observed SO$_2$ columns.

The a posteriori SO$_2$ emissions show substantial reductions during the years 2005-2018 over China (by -6.1%/yr for the
country total), some parts of Europe (by up to -6%/yr at grid scale), the eastern United States (up to -3%/yr) and Japan (up
to -8%/yr), whereas it shows strong increase over India (up to 5%/yr), the Middle East (up to 4%/yr), and Mexico (about 4%
around Mexico city). The negative trends are particularly large over central and southwestern China, which is due to strong
emission reductions after 2010 (Fig. S8), as reported by Li et al. (2017) and Koukouli et al. (2018). In contrast, the reductions
are smaller for northwestern China, which could be attributed to the exceptional positive trend in this region after 2010 (Ling
et al., 2017). The obtained strong emission changes (summarized in Table S5), along with changes in NO$_x$ emissions, have
strong implications into the secondary aerosol formation processes for many polluted regions.

The a posteriori SO$_2$ emissions seem excessively high in 2011 for many regions (c.f., Fig. S8), which seems unrealistic
and could be due to potential problems in data assimilation setting or assimilated retrievals. Volcanic eruptions also affected a
temporal increase in the estimated emissions, as shown by Carn et al. (2017) using the OMI SO$_2$ measurements. This requires
additional careful verification before used in detailed trend analysis. The estimated emissions should have a large uncertainty
associated with large retrieval uncertainty (e.g., random noise of 0.5 DU for remote areas, as described in Li et al. (2013)) and
the assumed constant retrieval errors and air mass factor. Because the optimized emission factors were applied to the a priori
emissions, any missing sources in a priori inventories (e.g. Liu et al., 2018) could also lead to systematic biases in the estimated
emissions.

## 5.4 Lightning NO$_x$ sources

The multi-constituent data assimilation with different vertical sensitivities provides strong constraints to distinguish between
surface and lightning NO$_x$ sources and to correct the vertical profiles of lightning NO$_x$ sources. The a posteriori global total
lightning NO$_x$ source is 7.5 TgN, which is about 32% higher than in the control run (5.7 TgN). The estimated global total
emission is about 17% larger than our previous estimates (6.4 TgN) based on the TCR-1 system (Miyazaki et al., 2014).
The differences between two estimates can primarily be attributed to change in the forecast model and its resolution. The
resolution improvement affected the representation of cumulus convection and lightning frequency distributions. Nevertheless,
both estimates suggest common problems of the lightning parameterizations such as requirements to modify the C-shape
assumption and land-ocean contrasts.





The long-term trends of lightning $NO_x$ are mostly insignificant and dominated by multi-year scale variability rather than linear increase or decrease (Fig. S9 and Table S6). The inter-annual variability of lightning $NO_x$ during 2005-2018 is large over Southeast and South Asia, central and southern Africa, Central Africa, and the Amazon (Fig. 14). Over central Africa, the lightning $NO_x$ sources are large in 2006 and 2008 and small in 2016. The lightning $NO_x$ sources also show strong interannual variations over the Amazon, with a maximum in 2009 and minimum in 2015. These changes are considered to be connected with climate variability such as ENSO (Rowlinson et al., 2019), associated with variations in convective activity, thunderstorm type, and cloud distributions. The lack of TES ozone measurements after 2010 introduced artificial changes, whereas the variations are considered to be consistent during 2005-2009 and 2010-2018 in the reanalysis when the observation density is nearly constant. Further detailed analyses are required to understand the possible causal mechanisms of the multi-year variability, which would provide important implications into chemistry-climate interaction processes. The regional total values of the estimated lightning NOx sources for major source regions.

## 6   Trend diagnostics

The reanalysis reveals substantial changes in concentrations of various species, which provides an important framework to comprehend the roles of natural and human activities on atmospheric composition. We evaluated long-term atmospheric composition variations using two data sets: the standard reanalysis products and a reanalysis without TES measurements (noTES reanalysis). The two data sets are identical after 2010, whereas in the standard reanalysis corrections made by the TES measurements for 2005-2009 could lead to artificial decadal trends during the reanalysis. The noTES reanalysis is meant to provide a consistent long-term record. As shown in Figs. 16, S10 and S11, the noTES reanalysis reveals positive trends for the surface ozone over many regions, with strong positive trends over India (up to 0.25 ppb/yr), Southeast Asia (up to 0.4 ppb/yr), and over the northern Pacific (up to 0.3 ppb/yr). Positive trends for surface ozone are also found throughout the SH. In contrast, strong reductions appear over the United States (up to -0.2 ppb/yr) and Europe (up to -0.15 ppb/yr). At 500 hPa (Figs. 16, S12 and S13), the linear ozone trends are overall positive except around the equator. The positive trend at 500 hPa reaches 0.3-0.4 ppb/year over the SH tropics, the tropical Atlantic, and the Middle East in the noTES reanalysis, which were mostly attributable to changes in anthropogenic $NO_x$ emissions. The strong increasing surface ozone trends indicate strong impacts of human activity on air quality, human health and climate over the past decade. The increases in the extratropical UTLS region can be driven by changes in STEs. The standard reanalysis products exhibit large positive trends at low latitudes and negative trends over most of the extratropics, associated with systematic biases between the model and TES measurements during 2005-2009.

The estimated global tropospheric ozone burden in the noTES reanalysis was $330.6 \pm 5.8$ Tg for 2005-2018, which is comparable to the 15-model mean value of 337 Tg from the Atmospheric Chemistry and Climate Model Intercomparison Project (ACCMIP) for 2000 (Young et al., 2013) and is slightly larger than the estimates of 300 Tg from the five satellite products for the years 2014–2016 (Gaudel et al., 2018), which could be partly attributed to the limited sensitivity of the satellite measurements to the lower troposphere and polar regions. The noTES reanalysis revealed a slight increase in global tropospheric





ozone burden (+0.4 Tg/yr) during 2005-2018. Because of the corrections by TES measurements, the global tropospheric ozone burden was 3.5 % lower in the reanalysis (317.0 Tg) than in the noTES reanalysis (328.7 Tg) for 2005-2009, which is closer to the satellite-based estimates.

According to changes in concentrations of various species including ozone, the reanalysis reveals a general positive trend in OH during the reanalysis period (Figs. 17, S3 and S4). The tropospheric OH from the noTES reanalysis exhibits strong increases over the tropical western and eastern Pacific by up to +1.2%/year, and 0.9-1.4%/yr over southern India, southern Vietnam, west coast of Saudi Arabia, and western Iran. Annual and zonal mean OH concentrations in the noTES reanalysis are increased over 10°N-20°N, 700-500 hPa by 0.5-0.6%/yr and at the SH low and mid-latitudes in the lower troposphere by 0.3-0.4%/yr. These trends are commonly found in both data sets, but with weaker trends in the standard reanalysis. At the NH mid latitudes in the free troposphere, only the noTES reanalysis reveals substantial increases in OH by 0.5-0.7%/year. Based on a sensitivity calculation, these significant changes in OH were found to be strongly driven by surface $NO_x$ emission variations, with strong increases from 2007 to 2012. These results highlight substantial impact of human activity on the oxidation capacity of the atmosphere and chemical lifetime of many species such as methane (e.g. Rigby et al., 2017), as previously suggested by Wang and Jacob (1998).

## 7 Discussions

### 7.1 Assimilated data biases and availability

Significant temporal changes in the reanalysis quality can partly be attributed to discontinuities in the observing systems. As discussed in Section 6, the reduced number of assimilated TES ozone retrievals after 2010 substantially influenced the usability of the reanalysis products for trend analyses. Meanwhile, changes in the $NO_2$ observing system, including the OMI row anomaly after December 2009 and the limited temporal coverage of SCIAMACHY and GOME-2, are also considered to affect long-term consistency. The reanalysis ozone bias against the ozonesonde measurements was relatively large in the tropical lower and middle troposphere, which could partly be attributed to the positive biases in the assimilated TES measurements. Miyazaki et al. (2015) tested a bias correction scheme for assimilation of TES ozone based on evaluation results using ozonesonde measurements (Boxe et al., 2010; Verstraeten et al., 2013), however the results were not always positive because of the difficulty in estimating the detailed bias structure. The reanalysis ozone bias can also be affected by biases in ozone precursors measurements such as $NO_2$ measurements. Nevertheless, we did not apply any bias correction to any assimilated measurements in the reanalysis, because of the difficulty in estimating the bias structure including inter-measurement biases. To improve the temporal consistency, a detailed assessment of biases in individual retrievals (e.g. Compernolle et al., 2020) and between different retrieval products would be helpful, as already tested in the CAMS reanalysis (Inness et al., 2019)

The availability of the ozonesonde measurements for the most recent years was also limited at the time of this research, which limits the evaluation of the reanalysis performance. The mean ozonesonde concentrations at SH mid latitudes show rapid changes after 2017, which were associated with the reduced number of ozonesonde observations. The current ozonesonde network is also too sparse to capture the regional and monthly representative ozone fields especially in the tropics, which can





lead to substantial sampling biases in the reanalysis performance, as discussed for evaluations of chemistry-climate models (Miyazaki and Bowman, 2017).

## 7.2 Impact of forecast model performance

Even though the assimilation of multi-species data influences the representation of various chemical fields including precursor emissions, the remaining model errors, such as chemical reaction rates and deposition rates, as well as meteorology, limit the data assimilation improvements. Miyazaki et al. (2020) developed a MOMO-Chem framework using four global CTMs and an EnKF data assimilation that directly accounts for model error in transport and chemistry. They demonstrated that the observational density and accuracy was sufficient for the assimilation to reduce the influence of model errors in data

assimilation analysis; i.e., multi-model spread of ozone analysis is reduced by 20-85% in the free troposphere. Model negative biases in tropospheric $NO_2$ column and surface CO in the NH are also greatly reduced by more than 40% in all models. MOMO-Chem provides integrated unique information on the tropospheric chemistry system and its uncertainty ranges, which would benefit future development of chemical reanalysis.

   Meanwhile, a strong reanalysis dependence on forecast model performance was found on the near surface concentrations

and precursor emissions, associated with insufficient observational constraints (Huijnen et al., 2019; Miyazaki et al., 2020). The ozone response to precursor's emissions was also found to be strongly sensitive to the chemical mechanisms in the model, which varied by a factor of 2 for end-member models, revealing fundamental differences in the representation of fast chemical and dynamical processes (Miyazaki et al., 2020). The emissions of ozone precursors other than $NO_x$ and CO, such as VOCs, have a pronounced influence on the tropospheric chemistry. Adjusting additional model parameters such as VOC emissions,

deposition, and/or chemical reactions rates could help reduce model errors. Furthermore, a simultaneous assimilation of trace gas and aerosol measurements would also reduce systematic model errors and provide more comprehensive information on various applications. Meanwhile, high-resolution modeling is also essential for accurate modeling of non-linear chemistry and resolving rapid variations in air pollutions and emissions around cities (Valin et al., 2011; Sekiya et al., 2018), which is also needed to improve the reanalysis performance.

## 745   7.3 Challenges with next generation satellite data

Next generation satellite data products, that have improved vertical sensitivity and accuracy, as well as improved spatial sampling, have great potential to further improve emissions and surface ozone analyses. The exploitation of existing sounders and development of multispectral retrievals is expected to add constraints on the reanalysis and to remove remaining model errors. For instance, as demonstrated in Section 3.2.2, the multispectral AIRS/OMI ozone retrievals provide decadal records of

tropospheric ozone. Miyazaki et al. (2019b) demonstrated that assimilation of AIRS/OMI ozone data, together with precursors and stratospheric measurements, improved the tropospheric ozone analysis over East Asia during the KORUS-AQ campaign for any meteorological conditions. This would provide important constraints on the decadal ozone variations in the reanalysis.

   Tropospheric PAN retrievals from TES were used to evaluate the reanalysis fields over both polluted and remote regions (c.f., Section 4.1.2). Cross-Track Infrared Sounder (CrIS) on Suomi-NPP also provides tropospheric PAN retrievals with im-





proved coverage and accuracy compared with TES (Payne et al., 2019). Assimilating PAN retrievals from TES and CrIS can be expected to improve the representation of the global nitrogen cycles, which would also benefit surface and lightning $NO_x$ emission estimates combining with tropospheric $NO_2$ column measurements. Meanwhile, TROPOMI provides global maps of the tropospheric $NO_2$ column on a daily basis with improved accuracy and higher spatial resolution compared with OMI (Griffin et al., 2019). Assimilating TROPOMI $NO_2$ has potential for improved evaluation of the changing landscape of emis-

sions on urban-to-regional and regional-to-global scales (Lorente et al., 2019). Assimilation of other retrievals such as OMI and TROPOMI $CH_2O$, CrIS Isoprene (Fu et al., 2019), and TES, CrIS, and IASI $NH_3$ (Shephard and Cady-Pereira, 2015) would also help improve the model chemistry and tropospheric ozone reanalysis.

### 7.4  Under-constrained CO emissions

The validation results of CO concentrations suggested under-corrected surface emissions of CO, especially in the NH extratrop-

ics (c.f., Section 4.3). There are several reasons for this. First, while our previous estimates in TCR-1 used MOPITT TIR-only CO profile data at 700 hPa, TCR-2 used TIR/NIR total column retrievals. The truly optimal settings of data assimilation parameters probably differ between the two setups. The TCR-2 setting might require further optimization. Second, the chi-square and observation-minus-forecast statistics suggested underestimated background errors of CO for many regions. Considering different systematic model errors and the increased model resolution between TCR-1 and TCR-2, background error inflation

settings need to be further optimized for TCR-2. Third, the data assimilation window (two-hour) used is clearly too short for CO emission estimates, considering its relatively long chemical lifetime and the coverage and limited near-surface sensitivity of MOPITT measurements. A longer data assimilation window for CO emission estimates, while keeping the short window for short-lived species such as $NO_x$ and ozone, would be required. Finally, CO is produced by the oxidation of methane and biogenic NMHCs (Duncan et al., 2007). These components can account for part of the missing CO concentrations. Adding more

observational constraints, such as for $CH_2O$ and methane, would help improve CO emission estimates (e.g. Stein et al., 2014; Zheng et al., 2018). We have already tested some of the developments and obtained improved estimates of CO concentrations and emissions, which will be implemented in the next generation chemical reanalysis.

### 7.5  Uncertainty estimation

Important information regarding the reanalysis product is provided by the error covariance. Within the EnKF assimilation

framework, the analysis ensemble spread is estimated from the standard deviation across the ensemble and provides a measure of the uncertainty of the reanalysis product. The information on the analysis uncertainty is included in the TCR-2 reanalysis products. For instance, as shown in Fig. S14, the analysis spread for ozone is about 1-3 ppb in the tropics and subtropics and 3-12 ppbv in the extratropics before 2011. These variations may be related to spatial variations in observation errors, the number of assimilated measurements, and model errors. After 2011, the spread mostly becomes smaller than 3 ppb for

the globe. The analysis uncertainty after 2011 seems excessively small as compared with the validation results against the ozonesonde measurements (c.f., Section 4.1.1), which is likely associated with the stiff tropospheric chemical system and lack of observational constraints. The obtained results indicate the requirements for additional observational information and/or




stronger covariance inflation for measuring the analysis spread corresponding to actual analysis uncertainty. At 200 hPa, the analysis spread is about 1-4 ppb in the tropical upper troposphere and about 20-80 ppb in the extratropical lower stratosphere.

The relative value (compared to the analysis ozone) is smaller in the extratropics because of the high accuracy of the MLS measurements. For other species, further investigations would be required to clarify the usefulness of the estimated uncertainty (i.e., analysis spread).

### 7.6 Implications for climate studies

The long-term reanalysis products allow detailed evaluations of inter-annual and decadal variations in atmospheric composi-
tion simulated by chemistry-climate and chemistry-transport models in association with changes in human activities and natural processes. Employing TCR-1, Miyazaki and Bowman (2017) evaluated the ACCMIP tropospheric ozone simulations and investigated sampling biases in model evaluation results when using the ozonesonde network. Evaluations of ozone simulations using chemical reanalyses provide important information on the performance of the simulated radiative forcing (e.g. Bowman et al., 2013; Stevenson et al., 2013; Kuai et al., 2020), attribution of radiative forcing (Bowman and Henze, 2012), and emergent
constraints on future projections (Miyazaki and Bowman, 2017; Bowman et al., 2018). Validation of short-lived species can be used to identify potential sources of error in model fields and is also important for evaluating the radiative forcing because simulated OH fields influence simulated climates through their influences on methane (Naik et al., 2013; Voulgarakis et al., 2013). The optimized precursor emission fields can be used to validate bottom-up emission inventories and lightning parameterizations. As changes in tropospheric ozone burden and $NO_x$ emissions show a close relation in different future scenarios
(Stevenson et al., 2013), evaluations using the estimated emissions and evaluated model response to emissions (Miyazaki et al., 2020) have the potential to evaluate preindustrial, present day, and future model simulations. Short-lived climate pollutants (SLCP) are an increasingly important component of greenhouse gas budgets that limit warming to target temperatures, e.g., 1.5C or 2C (Rogelj et al., 2019). Chemical reanalysis can play a crucial role in assessing the changes and efficacy of SLCPs.

## 8 Conclusions

We conducted a tropospheric chemical reanalysis calculation for the 14 years from 2005 to 2018 based on an assimilation of multi-constituent observations from multiple satellite sensors. The assimilated measurements of ozone, $NO_2$, CO, $HNO_3$, and $SO_2$ were obtained from the OMI, SCIAMACHY, GOME-2, TES, MLS, and MOPITT satellite instruments. Surface emissions of $NO_x$, CO, and $SO_2$ and lightning $NO_x$ sources and the chemical concentrations of various species are simultaneously optimized using an EnKF data assimilation. In this framework, the improved concentrations of various species have the poten-
tial to improve the emission inversion, whereas the improved representations of emissions benefit the concentration reanalysis through a reduction in the model errors.

The evaluation results for various species reveal the benefit of the assimilation of multiple-species data on the analysis of both observed and unobserved species profiles on both regional and global scales, for seasonal and decadal variations, and from the surface to lower stratosphere. The reanalysis ozone bias against the ozonesonde measurements was less than 1.2



ppb in the lower troposphere except for the tropics and less than 3.1 ppb in the middle and upper troposphere except for the SH high latitudes, with temporal correlations greater than 0.85 for most regions. The improved agreements in TCR-2 ozone from TCR-1 can be attributed to a mixture of various upgrades, including assimilated measurements and the forecast model performance and resolution. The assimilation also removed the global mean model biases in the tropospheric $NO_2$ column by about 84-93%, while reproducing the observed seasonal and inter-annual changes for both industrialized and biomass burning

regions (r = 0.88-0.99). The model biases in surface CO concentrations are greatly reduced in the SH, the tropics, and NH mid latitudes by 66-88%. The reanalysis also reasonably captured the observed spatial and temporal variability in PAN as compared with the TES satellite retrievals (r = 0.52-0.84 for the seasonal mean fields). The negative model biases (by 10-70%) in the free tropospheric PAN are greatly reduced by data assimilation compared to the aircraft measurements due to increased surface and lightning $NO_x$ emissions. Data assimilation also removed positive model biases for $SO_2$ in the lower and middle

troposphere. The reanalysis OH shows improved agreements in global distributions over remote oceans in comparison with the ATom aircraft measurements from the surface to the upper troposphere, with the RMSE reduction of up to 30% in the free troposphere and improved north-to-south gradients. Constraints obtained for OH profiles have a large potential to influence the chemistry of the entire troposphere, which played an important role in propagating observational information among various species and in modifying the chemical lifetimes of many species. Although no aerosol observations were assimilated, improved

representations of aerosols against surface in-situ measurements were obtained through corrections made to the secondary aerosol formation.

The multi-constituent data assimilation framework is also used to improve estimates of global emissions of $NO_x$, CO, and $SO_2$. The simultaneous optimization of emissions and concentrations reduces the model-observation mismatches that arise from model errors other than those related to emissions. The global total emissions averaged over the 14 years is estimated

at 49.2 TgN/yr for surface $NO_x$ emissions, 1104 TgCO/yr for surface CO emissions, 35.1 TgS/yr for surface $SO_2$ emissions, and 7.5 TgN/yr for lightning $NO_x$ sources, which are substantially different from the a priori emissions constructed based on bottom-up inventories. Chinese $NO_x$ emissions increased from 2005 to 2011, then rapidly decreased after 2013, and then started to increase since 2016, while exhibiting substantial spatial differences within the country. Indian $NO_x$ emissions exhibit a continuous increase by 30% over 14 years. For the United States and Europe, the $NO_x$ emissions show a slowdown in $NO_x$

emission reductions in the recent years. The $SO_2$ emissions show substantial reductions over China (by -6.1%/yr), some parts of Europe (up to -6%/yr on each grid), the eastern United States (up to -3%/yr) and Japan (up to -8%/yr), whereas strong increases are found over India (up to 5%/yr), the Middle East (up to 4%), and Mexico (about 4%), all of which are associated with environmental policies and economic activities. Lightning $NO_x$ sources exhibit strong year-to-year variability, associated with multi-year scale climate variability such as ENSO. The multi-year changes in emissions, along with the changes in

meteorological conditions, led to strong increases in surface ozone over India (up to +0.25 ppb/yr) and Southeast Asia (up to +0.4 ppb/yr), as well as in tropospheric OH over the tropical western and eastern Pacific (up to +1.2%/yr) and low latitudes polluted areas (0.9-1.4%/yr) during 2005-2018. These results have strong implications on the impacts of human activity on air quality, human health, and climate. Meanwhile, significant temporal changes in the reanalysis can partly be attributed to discontinuities in the observing systems.

The combined analysis of concentrations and emissions is considered an important development in the tropospheric chemistry reanalysis. Our comparisons suggest that improving the observational constraints, including the continued development of satellite observing systems, together with the optimization of model parameterizations, such as deposition and chemical reactions, will lead to increasingly consistent long-term reanalyses in the future. An increase in the forecast model resolution and an extension of data assimilation to aerosols are expected to improve the capability of chemical reanalysis for air quality

and climate applications. Techniques to reduce the influence of discontinuities in the assimilated measurements and to employ next generation satellite retrievals would also be important developments in future chemical reanalyses. Satellite data sets from a new constellation of LEO sounders and GEO satellites (e.g., GEMS, TEMPO and Sentinel-4) will provide more detailed knowledge of ozone and its precursors for East Asia (Bowman, 2013).

## 9   Data availability

The Tropospheric Chemistry Reanalysis (TCR-2) data for 2005-2018 is freely available at https://doi.org/10.25966/9qgv-fe81 (Miyazaki et al., 2019a). The teaser data (mon_emi_nox_tot_2005.nc) is a part of the TCR-2 data products (with the same DOI) and can be downloaded from the TCR-2 data website by selecting "Monthly-mean data: Emissions" - "NOx (surface total)" - "2005" at https://tes.jpl.nasa.gov/chemical-reanalysis/products/monthly-mean.

*Author contributions.*   KM and KB initiated the research. KM and TS conducted the TCR-2 calculations. TS, KS, MS, KO contributed to

the TCR-2 system developments. HE and FB provided OMI, SCIAMACHY, and GOME-2 $NO_2$ data. NL provided MLS ozone and $HNO_3$ data. HW provided MOPITT CO data. VHP provided input on the use of the TES PAN data. All authors contributed to the review and editing of the manuscript

*Competing interests.*   The authors declare that they have no conflict of interest.

*Acknowledgements.*   We acknowledge the use of data products from the NASA AURA and EOS Terra satellite missions. We also acknowl-

edge the free use of tropospheric $NO_2$ column data from the SCIAMACHY, GOME-2, and OMI sensors from www.temis.nl, NASA's aircraft observations data were obtained from the NASA Langley Research Center Atmospheric Science Data Center. This research has been supported by the EU FP7 project, Quality Assurance for Essential Climate Variables (QA4ECV), grant no. 607405. Surface aerosol observations from the CASTNET, EANET, and EMEP networks. This work was supported by the Post-K computer project Priority Issue 4. The Earth Simulator was used for simulations as Strategic Project. Part of this work was conducted at the Jet Propulsion Laboratory, California Institute

of Technology, under contract with the National Aeronautics and Space Administration (NASA).





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



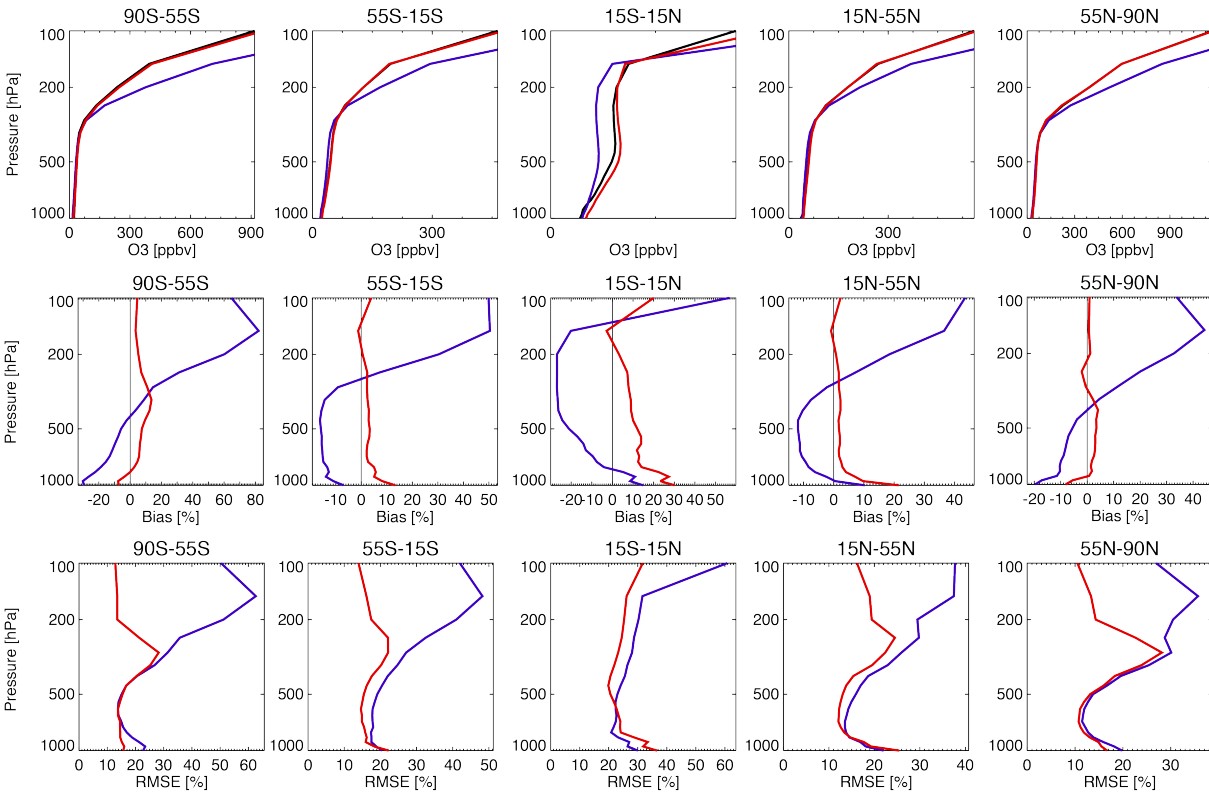

**Figure 1.** Comparison of the vertical ozone profiles between ozonesondes (black), control run (blue), and reanalysis (red) averaged for the period 2005–2018. The upper row shows the mean profile; center and lower rows show the mean difference and the RMSE between the control run and the observations (blue) and between the reanalysis and the observations (red). From left to right, results are shown for the SH high latitudes (55–90° S), SH mid latitudes (15–55° S), tropics (15° S–15° N), NH mid latitudes (15–55° N), and NH high latitudes (55–90° N).



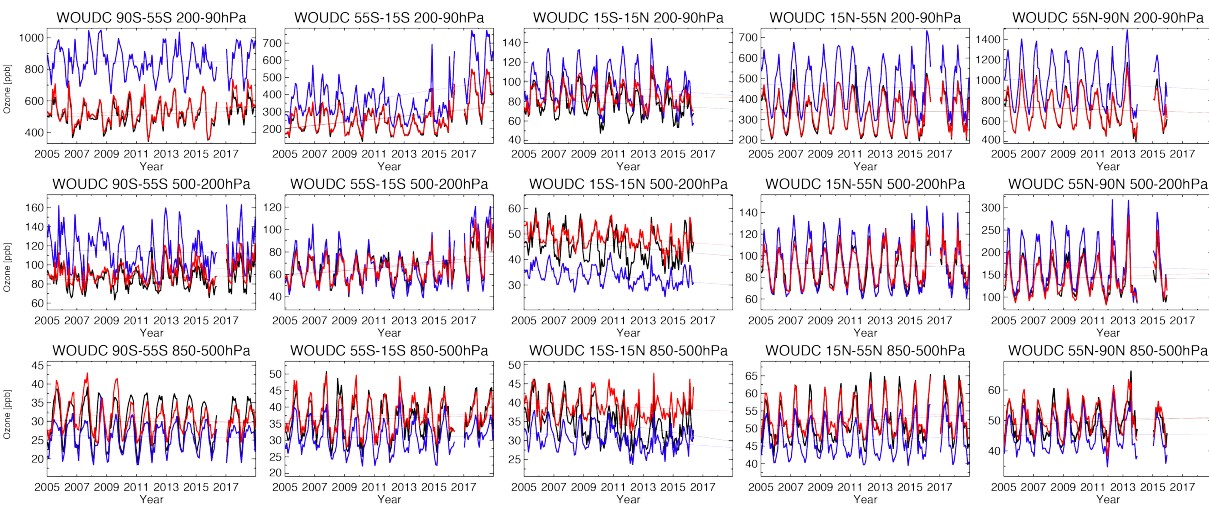

**Figure 2.** Time series of the monthly mean ozone concentration obtained from ozonesondes (black), control run (blue), and reanalysis (red) averaged between 850–500 hPa (upper row), 500–200 hPa (center row), and 200–90 hPa (lower row). From left to right the results are shown for the SH high latitudes (55–90° S), SH mid-latitudes (15–55° S), tropics (15° S–15° N), NH mid-latitudes (15–55° N), and NH high latitudes (55–90° N).

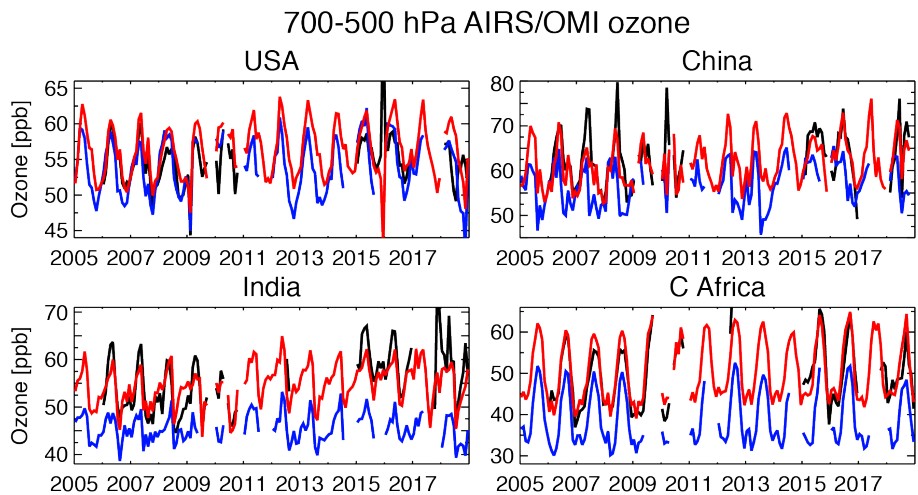

**Figure 3.** Time series of the monthly mean ozone concentration obtained from the AIRS/OMI retrievals (black), control run (blue), and reanalysis (red) averaged between 700–500 hPa over the United States (127–70° W, 28–50° N), India (68–89° N, 8–33° N), China (110–123° N, 30–40° N), and Central Africa (10–40° E, 20° S–Eq.).

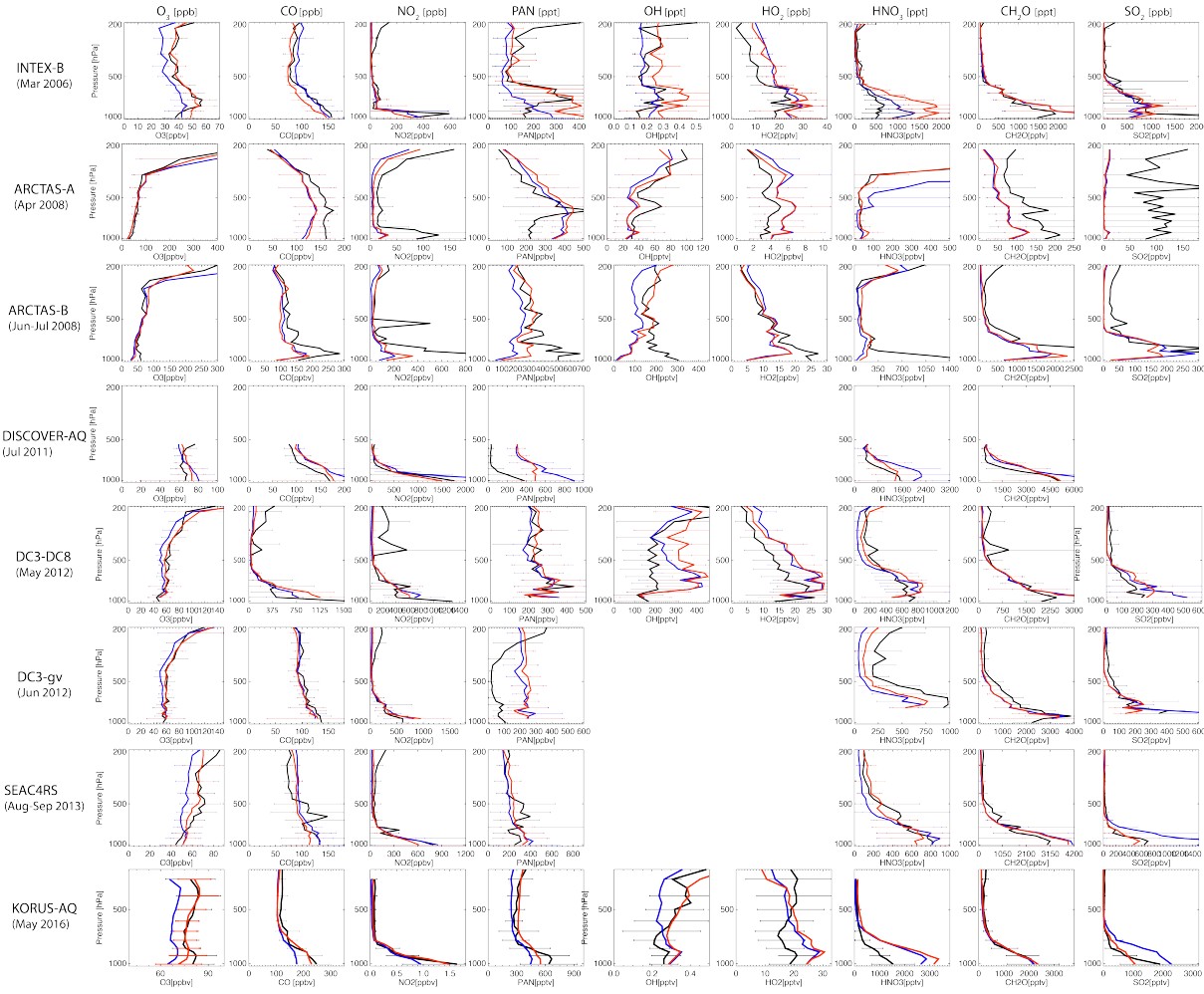

**Figure 4.** Mean vertical profiles of $O_3$ (ppb), CO (ppb), $NO_2$ (ppb), PAN (ppt), OH (ppt), $HO_2$ (ppb), $HNO_3$ (ppt), $CH_2O$ (ppt), and $SO_2$ (ppt) obtained from aircraft measurements (black), control run (blue), and reanalysis (red), for the INTEX-B profile (1st row), ARCTAS-A profile (2nd row), ARCTAS-B profile (3rd row), DISCOVER-AQ profile (4th row), DC3-DC8 profile (5th row), DC3-GV profile (6th row), SEAC4RC profile (7th row), and KORUS-AQ profile (8th row). Error bars represent the standard deviation of all data within one bin (with an interval of 30 hPa).



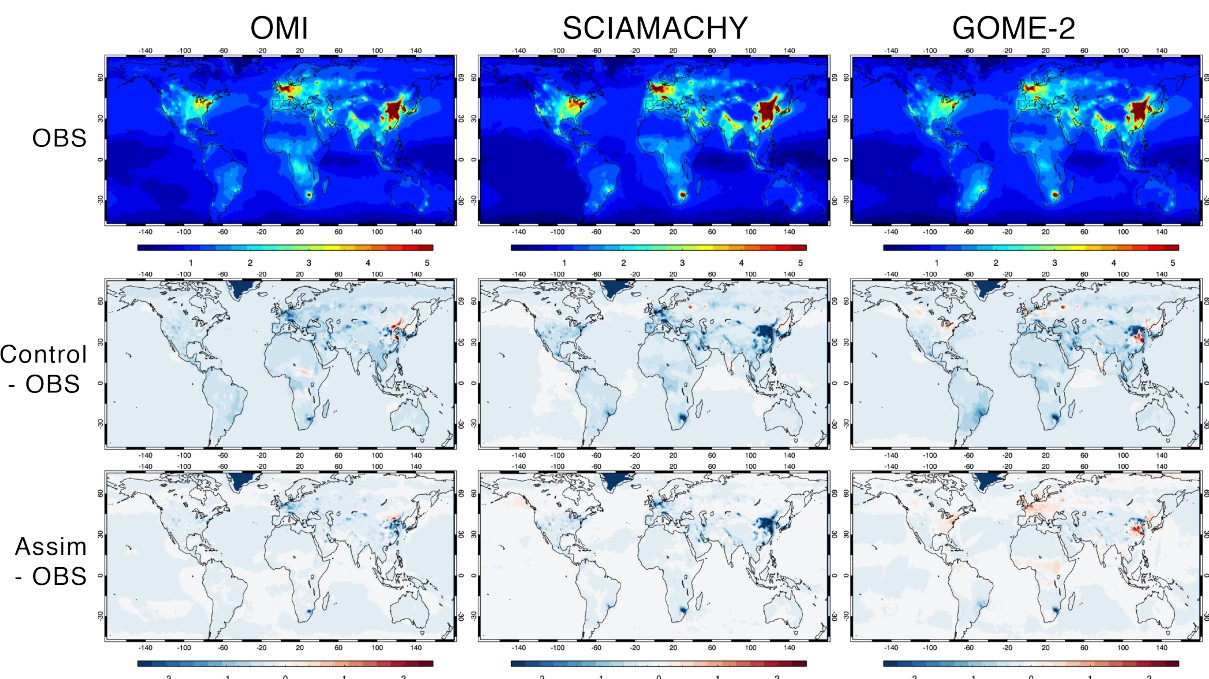

**Figure 5.** Global distributions of the tropospheric $NO_2$ columns (in $10^{15}\,\mathrm{molec\,cm^{-2}}$). The results are shown for OMI (left columns) for 2005-2018, SCIAMACHY (middle columns) for 2005-2011, and GOME-2 (right columns) for 2007-2018. Upper row shows the tropospheric $NO_2$ columns obtained from the satellite retrievals (OBS), centre row shows the difference between the model simulation and the satellite retrievals (Model-OBS); and lower row shows the difference between the data assimilation and the satellite retrievals (Assim-OBS).



**Figure 6.** Time series of regional monthly mean tropospheric $NO_2$ columns (in $10^{15}$ molec cm$^{-2}$) averaged over China (110–123°E, 30–40°N), Europe (10°W–30°E, 35–60°N), the United States (70–125°W, 28–50°N), India (68–89°E, 8–33°N), South America (50–70°W, 20°S–Equator), North Africa (20°W–40°E, Equator–20°N), Central Africa (10–40°E, Equator–20°S), Southern Africa (25–34°E, 22–31°S), Southeast Asia (96–105°E, 10–20°N), and Australia (113–155°E, 11–44°S) obtained from the satellite retrievals (black), control run (blue), and the data assimilation (red). Results are shown for the OMI retrievals (left columns), SCIAMACHY retrievals (centre columns), and GOME-2 retrievals (right columns).
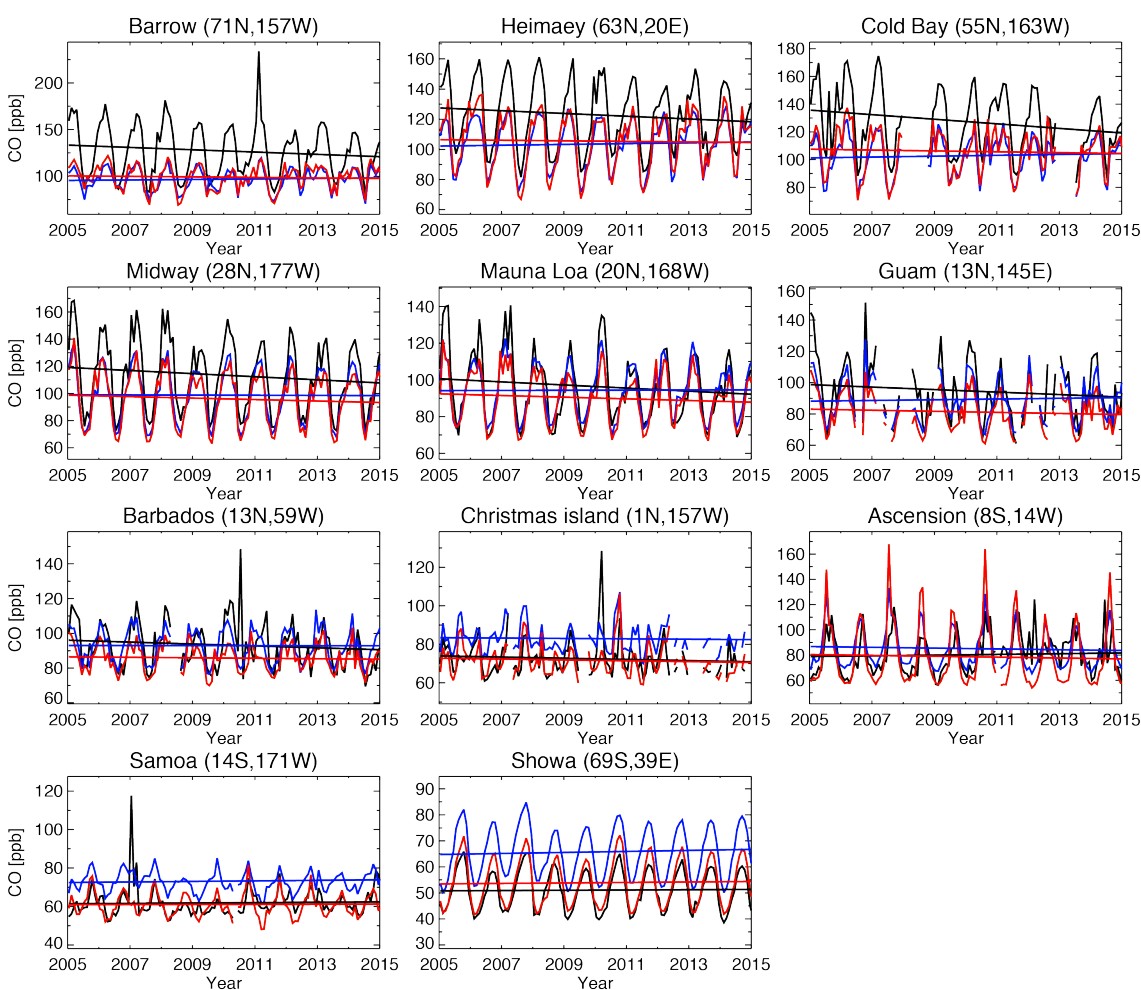

**Figure 7.** Time series of monthly mean surface CO concentrations obtained from the WDCGG ground measurements (black), control run (blue), and reanalysis (red).



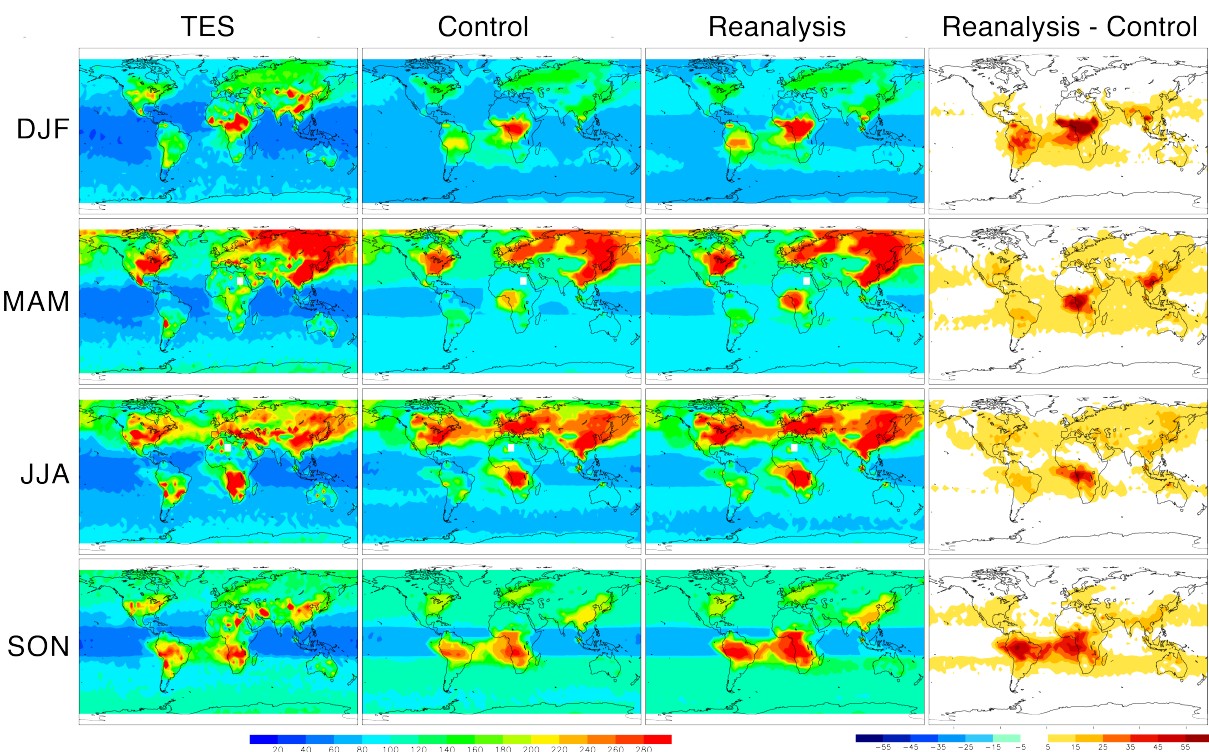

**Figure 8.** Global distributions of the PAN concentrations (in ppt) averaged between 800 and 400 hPa during 2005-2009. The results are shown for the TES retrievals (left columns), model simulation (2nd columns), reanalysis (3rd columns), and the reanalysis minus model (right columns) for DJF (top row), MAM (2nd row), JJA (3rd row), and SON (bottom row).



**Figure 9.** Latitude-pressure cross section of the mean OH concentration (right panels) averaged during 2005-2018 and time-latitude cross-section of the monthly mean OH concentration averaged between the surface and 300 hPa (left panels). The mean OH concentrations from the control run (upper row), reanalysis (center row) and the difference between the reanalysis and the model simulations (lower row) are shown. Units are $10^6$ molecules cm$^{-3}$.

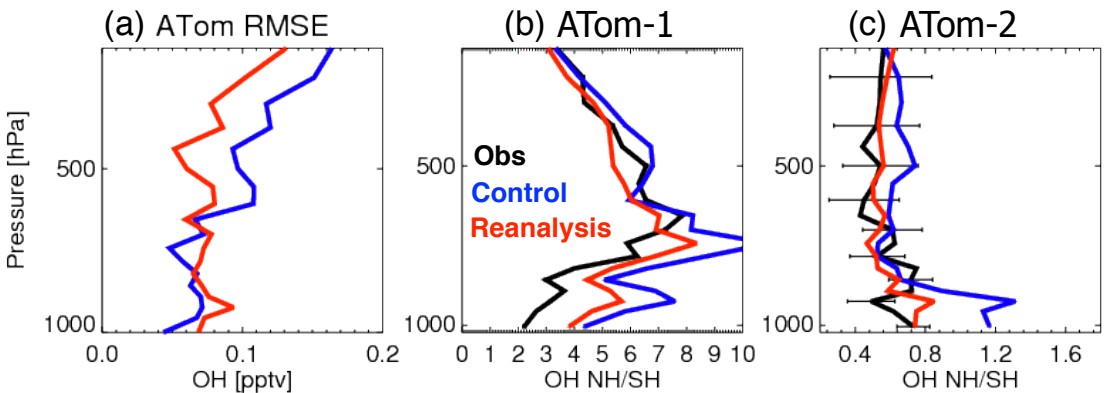

**Figure 10.** Vertical profiles of (a) OH RMSEs compared with the ATom-1 and ATom-2 aircraft observations (in pptv) and inter-hemispheric gradients of OH for (b) ATom-1 and (c) ATom-2 from the observations (black), control run (blue), and reanalysis (red).



**Figure 11.** Comparisons of monthly mean surface aerosol concentrations between the control run (blue) and reanalysis (red) with the EMEP (upper row), EANET (center row), and EMEP (lower row) observations for ammonium (left columns), nitrate (center columns), and sulfate (right columns) aerosols for 2005-2017.

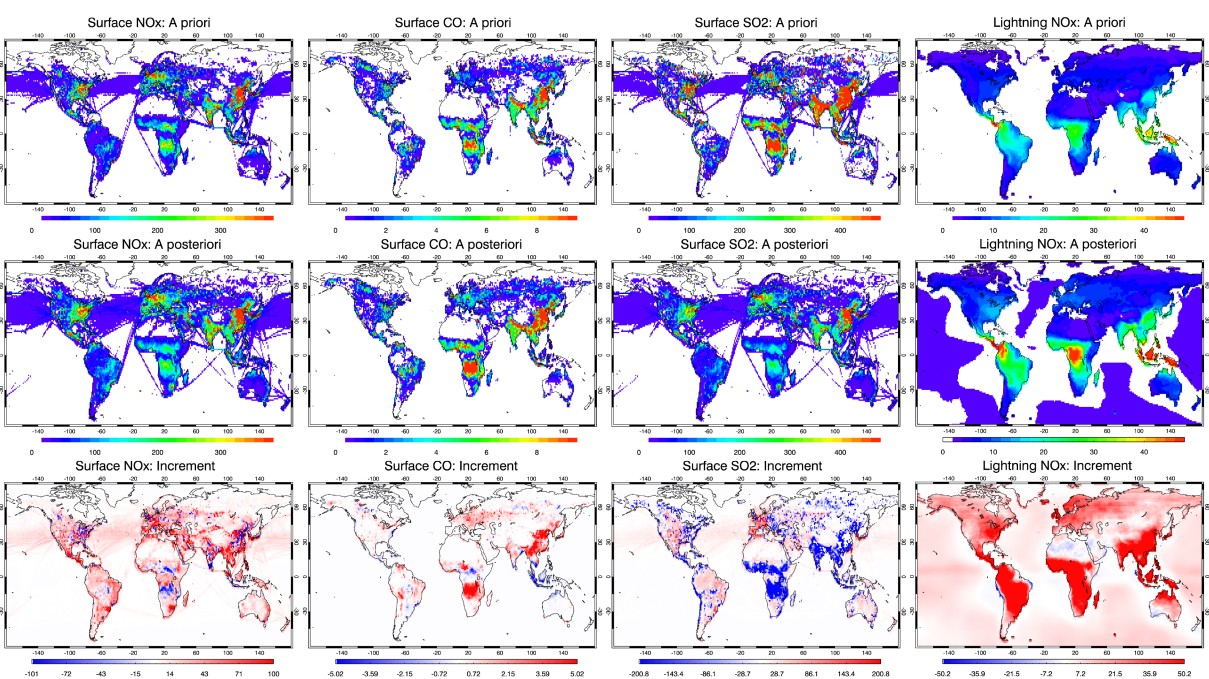

**Figure 12.** Global distributions of surface $NO_x$ emissions (in $10^{-13} kgNm^{-2}s^{-1}$) (left columns), surface CO emissions (in $10^{-10} kgCOm^{-2}s^{-1}$) (2nd columns), surface $SO_2$ emissions (in $10^{-13} kgSm^{-2}s^{-1}$) (3rd columns), and lighting $NO_x$ sources (in $10^{-14} kgNm^{-2}s^{-1}$) (right columns) averaged over 2005–2018. The a priori emissions (upper rows), a posteriori emissions (middle rows), and analysis increment (lower rows), i.e., the difference between the a posteriori and the a priori emissions, are shown for each panel.

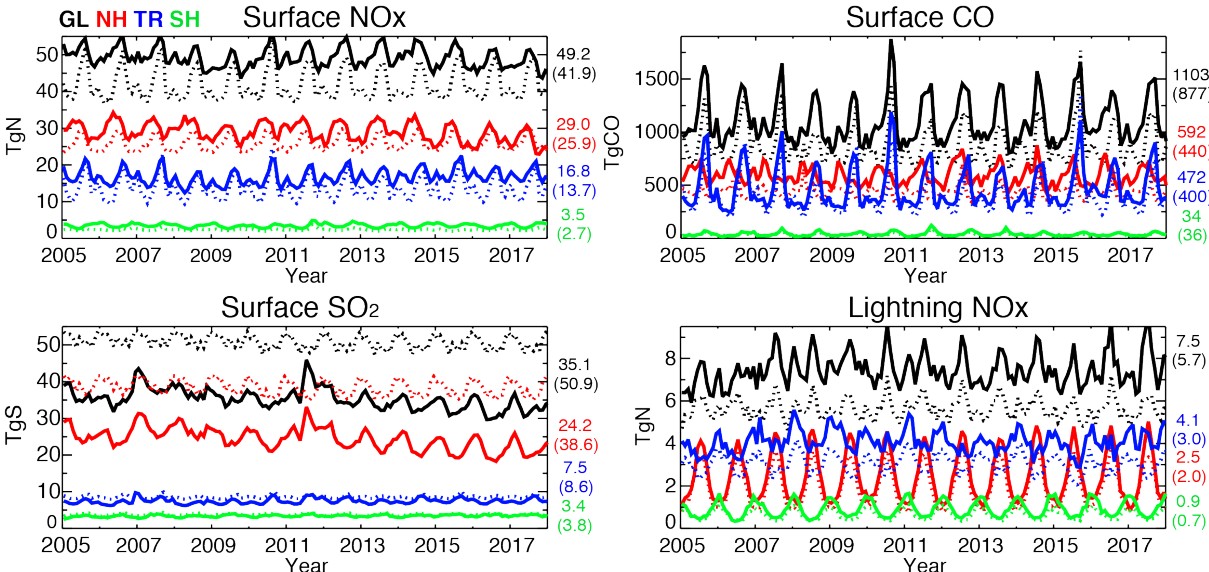

**Figure 13.** Time series of monthly total global and regional surface $NO_x$ emissions (in $Tg\,N\,yr^{-1}$), surface CO emissions (in $Tg\,CO\,yr^{-1}$), surface $SO_2$ emissions (in $Tg\,S\,yr^{-1}$), $LNO_x$ emissions (in $Tg\,N\,yr^{-1}$), and obtained from the reanalysis (solid lines) and the emission inventories or the control run (dashed lines) over the globe (90° S–90° N), NH (20–90° N), tropics (TR, 20° S–20° N), and SH (90–20° S). The mean emissions values obtained from the reanalysis run and the emission inventories (in bracket) averaged over the years 2005-2018 are shown on the right-hand side.



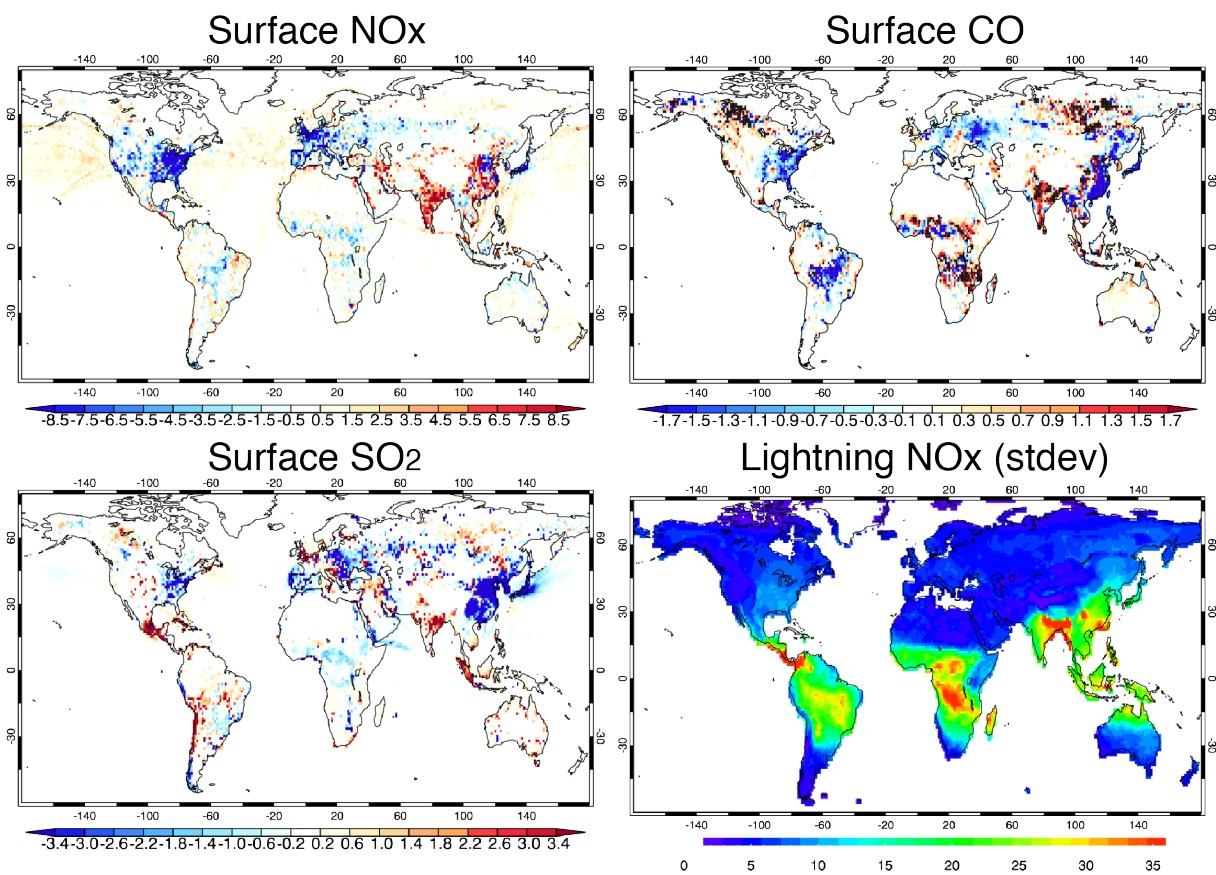

**Figure 14.** Global distribution of linear trend of the a posteriori surface $NO_x$ emissions (in $10^{-13}$kgNm$^{-2}$s$^{-1}$ per year), surface CO emissions (in $10^{-11}$kgCOm$^{-2}$s$^{-1}$ per year), and surface $SO_2$ emissions (in $10^{-14}$kgSm$^{-2}$s$^{-1}$ per year), and standard deviation of the a posteriori lightning $NO_x$ emissions (in $10^{-14}$kgNm$^{-2}$s$^{-1}$ per year) for the period 2005–2018. The red (blue) colour indicates positive (negative) trends.

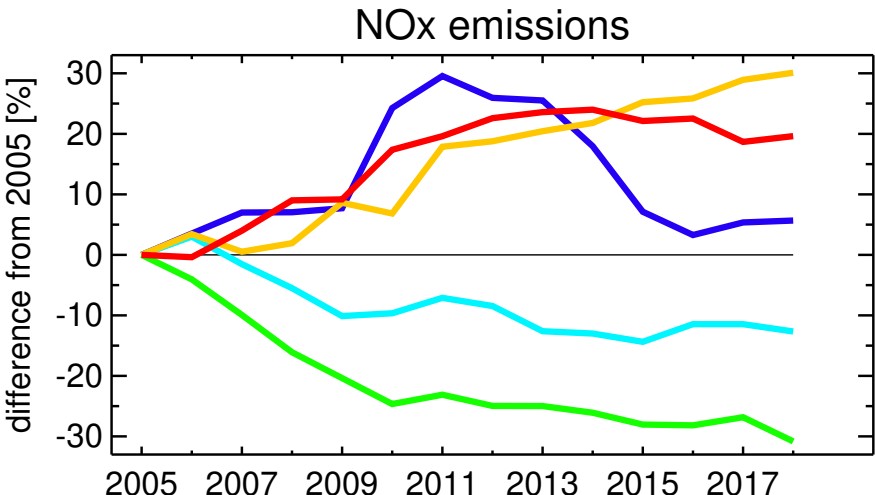

**Figure 15.** Time series of the difference (in %) of the annual mean a posteriori surface $NO_x$ emissions relative to the 2005 emissions in the period 2005–2018 for India (yellow), China (blue), Europe (light blue), the Middle East (red), and the United States (green).

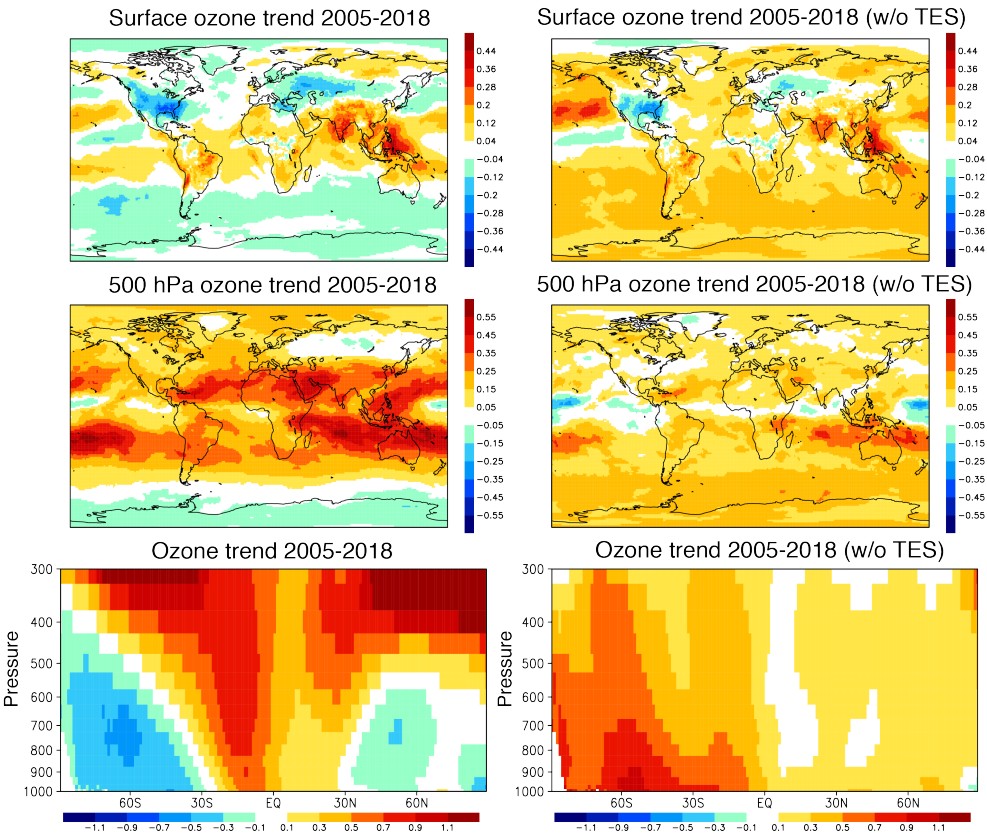

**Figure 16.** Global distribution of linear trend of ozone concentrations (in ppb per year) at the surface (upper row) and 500 hPa (center row) obtained from the reanalysis (left columns) and noTES reanalysis (right columns) for the years 2005-2018. The lower row shows latitude-pressure cross section of the linear trend (in % per year).



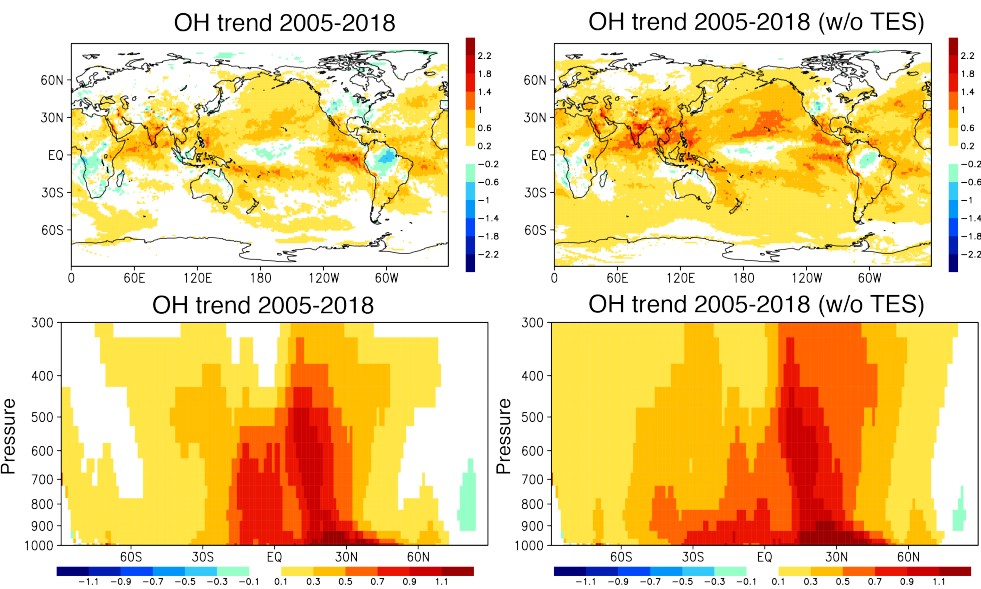

**Figure 17.** Same as in Fig. 16, but for tropospheric OH (in $10^6$ molecules cm$^{-3}$ per year, upper row) and the latitude-pressure cross-section of OH trends (in % per year, lower row) for the years 2005-2018.



**Table 1.** Comparisons of TCR-1 and TCR-2.

|  | TCR-1 (Miyazaki et al., 2015) | TCR-2 (This study) |
|---|---|---|
| Forecast model | AGCM-CHASER<br>47 species, 113 reactions | MIROC-CHASER<br>92 species, 262 reactions |
| Meteorological data | nudged to NCEP/DOE-II | nudged to ERA-Interim |
| State vector | NOx, CO emissions, lightning NOx, 35 chemical species | TCR-1 + $SO_2$ emissions |
| Assimilated data | OMI $NO_2$ (DOMINO2),<br>SCIAMACHY, GOME-2 $NO_2$ (TM4NO2A v2.3),<br>TES ozone (v5), MOPITT CO (v6 TIR),<br>MLS ozone, $HNO_3$ (v3.3) | OMI $NO_2$ (QA4ECV v1.1),<br>SCIAMACHY, GOME-2 $NO_2$ (QA4ECV v1.1),<br>TES O3 (v6), MOPITT CO (v7 TIR/NIR),<br>MLS ozone, $HNO_3$ (v4.2), OMI $SO_2$ (PCA) |
| A priori emissions | EDGAR v4.2, GFED v3.1, GEIA | HTAP v2, GFED v4, GEIA |
| Period | 2005-2012 | 2005-2018 |
| Resolution | 2.8°x2.8°, 32 layers to 4 hPa | 1.1°x1.1°, 32 layers to 4 hPa |



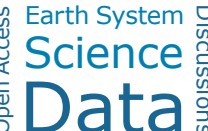

**Table 2.** Model and ozonesonde observation comparisons for the reanalysis and the control run (in brackets) for 2005-2018. The units of the root-mean-square error (RMSE) and bias are ppb.

|  |  | 90–55° S | | 55–15° S | | 15S–15° N | | 15–55° N | | 55–90° N | |
|---|---|---|---|---|---|---|---|---|---|---|---|
|  |  | Bias | RMSE | Bias | RMSE | Bias | RMSE | Bias | RMSE | Bias | RMSE |
|  | 850– | 0.9 | 4.1 | −0.4 | 5.3 | 4.2 | 7.8 | 1.0 | 6.7 | 1.2 | 5.8 |
|  | 500 | (-4.0) | (4.6) | (-6.2) | (6.8) | (-4.3) | (7.7) | (-5.4) | (7.5) | (-4.1) | (6.2) |
| WOUDC | 500– | 5.7 | 15.7 | -0.2 | 13.0 | 3.1 | 9.0 | 0.9 | 17.0 | -0.2 | 26.7 |
| sonde | 200 | (27.7) | (31.8) | (3.0) | (22.3) | (-11.9) | (12.2) | (2.6) | (22.4) | (27.0) | (39.1) |
|  | 200– | 22.2 | 67.5 | 3.4 | 42.7 | 7.3 | 22.0 | 1.9 | 59.1 | 2.6 | 86.0 |
|  | 90 | (332.7) | (261.6) | (142.7) | (126.9) | (16.6) | (37.0) | (125.1) | (120.9) | (256.4) | (211.3) |



**Table 3.** Comparisons of global tropospheric $NO_2$ columns between the control run and the satellite retrievals in brackets, and between the reanalysis run and the satellite retrievals: OMI for 2005–2018, SCIAMACHY for 2005–2011, and GOME-2 for 2007–2018. Shown are the global spatial correlation (S-Corr), the mean bias (BIAS: the data assimilation minus the satellite retrievals) and the root-mean-square error (RMSE) in $10^{15}\,\mathrm{molec\,cm^{-2}}$.

|        | OMI     | SCIAMACHY | GOME-2  |
|--------|---------|-----------|---------|
| S-Corr | 0.98    | 0.98      | 0.97    |
|        | (0.95)  | (0.96)    | (0.95)  |
| BIAS   | -0.03   | 0.01      | 0.02    |
|        | (-0.19) | (-0.15)   | (-0.20) |
| RMSE   | 0.17    | 0.27      | 0.24    |
|        | (0.30)  | (0.38)    | (0.38)  |



**Table 4.** The monthly mean bias and temporal correlation of regional mean tropospheric $NO_2$ columns: the data assimilation minus the satellite retrievals from OMI for the period 2005–2018 in $10^{15}\,\mathrm{molec\,cm^{-2}}$. The results of the model simulation (without data assimilation) are also shown in brackets.

|           | Bias    | T-Corr  |
|-----------|---------|---------|
| China     | -0.43   | 0.99    |
|           | (-0.34) | (0.92)  |
| Europe    | -0.23   | 0.95    |
|           | (-0.50) | (0.83)  |
| USA       | -0.12   | 0.88    |
|           | (-0.26) | (0.54)  |
| S-America | -0.01   | 0.98    |
|           | (-0.30) | (0.92)  |
| N-Africa  | 0.02    | 0.98    |
|           | (-0.24) | (0.93)  |
| C-Africa  | -0.01   | 0.99    |
|           | (-0.23) | (0.97)  |
| S-Africa  | -0.42   | 0.98    |
|           | (-0.80) | (0.94)  |
| SE-Asia   | -0.13   | 0.96    |
|           | (-0.60) | (0.88)  |
| Australia | -0.03   | 0.85    |
|           | (-0.22) | (0.85)  |
| India     | -0.04   | 0.96    |
|           | (-0.22) | (0.68)  |



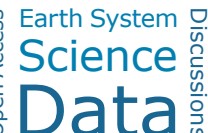

**Table 5.** Same as Table 2, but for surface CO concentrations. The unit is ppb. Observations used are the WDCGG observations during 2005–2014.

| 90–55° S | | 55–15° S | | 15S–15° N | | 15–55° N | | 55–90° N | |
|---|---|---|---|---|---|---|---|---|---|
| Bias | RMSE | Bias | RMSE | Bias | RMSE | Bias | RMSE | Bias | RMSE |
| 2.1 | 9.5 | 4.7 | 2.01 | -3.6 | 29.5 | 1.2 | 58.4 | -9.4 | 36.3 |
| (13.2) | (18.6) | (14.1) | (24.3) | (13.8) | (34.2) | (-9.4) | (50.2) | (-19.8) | (33.2) |





**Table 6.** Comparisons of surface aerosol concentrations between the control run and the in-situ observations in brackets, and between the reanalysis run and the in-situ observations for 2005-2017. Shown are the linear regression slope and intercept, the correlation (Corr), the mean and median biases, and RMSE in $\mu g\,m^{-3}$ for the EMEP, CASTNET, and EANET stations.

|  | Slope+intercept | Corr | Bias (mean) | Bias (median) | RMSE |
|---|---|---|---|---|---|
| EMEP | 0.47x+0.40 | 0.42 | -0.26 | 0.03 | 1.17 |
| Ammonium | (0.49x+0.43) | (0.27) | (-0.22) | (0.09) | (1.26) |
| CASTNET | 1.38x-0.10 | 0.93 | 0.19 | 0.14 | 0.30 |
| Ammonium | (2.14x-0.30) | (0.93) | (0.57) | (0.57) | (0.77) |
| EANET | 0.75x+0.41 | 0.62 | 0.12 | 0.16 | 0.78 |
| Ammonium | (0.98x+0.51) | (0.72) | (0.49) | (0.46) | (0.87) |
| EMEP | 0.26x+0.60 | 0.22 | -1.84 | -0.50 | 4.74 |
| Nitrate | (0.23x+0.55) | (0.20) | (-2.00) | (-0.59) | (4.81) |
| CASTNET | 1.89x-0.12 | 0.73 | 0.63 | 0.41 | 1.01 |
| Nitrate | (1.96x-0.21) | (0.71) | (0.60) | (0.31) | (1.03) |
| EANET | 0.70x+0.29 | 0.53 | -0.25 | -0.17 | 1.61 |
| Nitrate | (0.62x+0.02) | (0.43) | (-0.66) | (-0.53) | (1.82) |
| EMEP | 0.54x+0.34 | 0.44 | -0.65 | -0.37 | 1.40 |
| Sulfate | (0.73x+0.18) | (0.30) | (-0.42) | (-0.17) | (1.47) |
| CASTNET | 0.92x-0.10 | 0.85 | -0.43 | -0.68 | 0.65 |
| Sulfate | (2.12x-0.21) | (0.86) | (1.02) | (1.16) | (1.61) |
| EANET | 0.88x+0.29 | 0.74 | -1.68 | -1.42 | 1.65 |
| Sulfate | (1.00x+0.51) | (0.80) | (-0.17) | (0.08) | (2.66) |



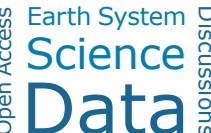

**Table 7.** The global and regional mean surface $NO_x$ (in Tg N yr$^{-1}$), CO (in Tg CO yr$^{-1}$), and $SO_2$ emissions (in Tg S yr$^{-1}$) and lightning $NO_x$ sources (in Tg N yr$^{-1}$) obtained from the a priori emissions (in brackets) and a posteriori emissions for the period 2005-2018. The results are shown for the Northern Hemisphere (NH, 20–90°N), the tropics (TR, 20°S–20°N), the Southern Hemisphere (SH, 90–20°S), and the globe (GL, 90°S–90.

|  | Globe | NH | TR | SH |
|---|---|---|---|---|
| Surface NOx | 49.2±2.8 | 29.0±2.6 | 16.8±2.1 | 3.5±0.5 |
|  | 41.9 | 25.9 | 13.7 | 2.7 |
| Surface CO | 1103.8±223.2 | 591.7±90.9 | 472.1±195.5 | 34.0±18.7 |
|  | (876.7) | (439.9) | (400.4) | (36.3) |
| Surface SO$_2$ | 35.1±3.0 | 24.2±3.1 | 7.5±0.7 | 3.4±0.3 |
|  | (50.9) | (38.6) | (8.6) | (3.8) |
| Lightning NO$_x$ | 7.5±0.8 | 2.5±1.2 | 4.1±0.5 | 0.9±0.4 |
|  | (5.7) | (2.0) | (3.0) | (0.6) |





**Table 8.** The regional mean surface $NO_x$ (in $Tg\,N\,yr^{-1}$), CO (in $Tg\,CO\,yr^{-1}$), and $SO_2$ emissions (in $Tg\,S\,yr^{-1}$) obtained from the a priori emissions and a posteriori emissions for the period 2005-2018 and their linear trends (in % per year). The results are shown for China (110–123°E, 30–40°N), Europe (10°W–30°E, 35–60°N), the United States (70–125°W, 28–50°N), South America (50–70°W, 20°S–Equator), North Africa (20°W–40°E, Equator–20°N), Central Africa (10–40°E, Equator–20°S), Southern Africa (25–34°E, 22–31°S), Southeast Asia (96–105°E, 10–20°N), Australia (113–155°E, 11–44°S), and India (68–89°E, 8–33°N).

| | China | Europe | US | S. America | N. Africa | C. Africa | S. Africa | SE Asia | Australia | India |
|---|---|---|---|---|---|---|---|---|---|---|
| NOx TCR-2 | 6.1 | 4.6 | 5.3 | 1.2 | 3.2 | 2.9 | 0.7 | 0.6 | 1.6 | 3.3 |
| NOx prior | 6.1 | 3.6 | 5.1 | 0.9 | 2.7 | 2.8 | 0.5 | 0.4 | 1.1 | 3.3 |
| NOx trend | 0.2 | -1.3 | -2.7 | -1.0 | -0.1 | 0.1 | 0.1 | 0.4 | 0.1 | 2.2 |
| CO TCR-2 | 176.9 | 42.4 | 64.6 | 38.5 | 107.1 | 164.9 | 7.9 | 13.7 | 15.4 | 78.0 |
| CO prior | 128.4 | 26.5 | 56.9 | 31.4 | 93.4 | 100.0 | 6.2 | 14.7 | 16.4 | 71.7 |
| CO trend | -0.6 | -0.8 | -1.8 | -4.7 | 0.8 | 0.9 | 0.8 | 0.1 | 1.8 | 1.5 |
| $SO_2$ TCR-2 | 5.8 | 2.8 | 2.5 | 0.4 | 0.6 | 0.7 | 0.5 | 0.1 | 1.2 | 1.8 |
| $SO_2$ prior | 11.6 | 3.8 | 5.8 | 0.2 | 0.5 | 0.7 | 1.1 | 0.4 | 1.6 | 5.0 |
| $SO_2$ trend | -6.1 | -1.1 | -1.1 | 2.1 | -1.9 | -0.4 | 2.0 | -0.1 | 0.9 | 2.2 |



**Table 9.** Same as in Table 7, but for lightning NOx sources $NO_x$ (in Tg N yr$^{-1}$) for North America (120–65° W, 20–60° N), Europe (10° W–30° E, 35–60° N), northern Eurasia (60–130° E, 30–68° N), the Pacific (154–180° E, 35° S–20° N and 180° E–88° W, 35° S–12° N), South America (77–39° W, 35° S–10° N), the Atlantic ocean (35° W–8° E, 30° S–3° N), northern Africa (15° W–48° E, 3–25° N), southern Africa (10–48° E, 30° S–3° N), the Indian ocean (52–108° E, 40–9° S), Southeast Asia (95–146° E, 9° S–26° N), and Australia (112–154° E, 40–12° S).

|            | Europe | N. America | S. America | S. Africa | N. Africa | Siberia | India | SE Asia | Pacific | Atlantic | Australia |
|------------|--------|------------|------------|-----------|-----------|---------|-------|---------|---------|----------|-----------|
| LNOx TCR-2 | 0.18   | 0.48       | 1.12       | 0.79      | 0.72      | 0.56    | 0.06  | 0.96    | 0.24    | 0.02     | 0.27      |
| LNOx prior | 0.13   | 0.39       | 0.87       | 0.56      | 0.55      | 0.47    | 0.03  | 0.72    | 0.12    | 0.02     | 0.23      |