# Peer review of "An updated tropospheric chemistry reanalysis and emission estimates, TCR-2, for 2005-2018"

_Earth System Science Data, 2020_

## Referee Comment (RC1) · Anonymous Referee #1 · 10 May 2020

This paper presents the results from the TCR-2 for the period 2005-2018 obtained from the assimilation of multiple updated satellite measurements. The derived emissions are validated against independent aircraft, satellite, and ozonesonde observations. The results are sound. I recommend the publication after minor revision.

General comments: 1. Section 2.2. The covariance matrices of observation and background error are very important for the accuracy of forecasts. Please clarify how the uncertainties are determined in the text. 2. Section 3.1. The super-observation approach is applied to $NO_2$ and CO, but not other species. Is there any specific reason for this? 3. Page 12, line 340, the authors tabulate the bias of modeled $NO_2$ compared to OMI, SCIAMACHY, and GOME-2 observations. I notice that the consistency is significantly better for results relying on single species (e.g., DECSO in Ding et al.).

[Figure]

What could be the driver for the bias? Is the uncertainty of the assimilated system or the conflict caused by satellite observations of different species used for assimilation? Line 357, Over India, the model negative bias increased with year because of the lack of the emission increases in the a priori emissions. I suggest adding some analysis about the uncertainty associated with the a priori emissions in the text as well. 4. Section 7.2. As mentioned by the authors VOCs have a pronounced influence on the tropospheric chemistry. Any specific reason for excluding satellite-observed VOC into the assimilated system?

Specific comments: 1. Page 3, line 58, the description of IFS and the following discussion about reanalysis pop up here. What is the relationship between TCR and those models? 2. Page 21, line 625, the negative increments over India seems to be inconsistent with the recent report. please try to clarify the reason for such inconsistency. 3. Fig 8 & 9, the legend is too small to see. 4. Fig 11, I recommend adding the r and RMSE in the figures. 5. The authors performed comprehensive evaluations for multiple species using multiple measurements. It is very impressive, but easy for readers to get lost. I recommend a table to summarize the validation results by species and sources of measurements. It would be easier for readers to follow the improvement by listing the global average difference between aprior and post simulations against measurements.

---

## Referee Comment (RC2) · Maarten Krol (Referee) · 17 May 2020

This paper presents a re-analysis of the tropospheric composition for the period 2005-2018. This re-analysis is constructed by assimilating a wide range of satellite observations in a high-resolution chemistry-transport model.

This is a great effort that should find its spin-off in the community. The paper is based on a large number of underlying studies that outline the method in more detail. This makes it difficult to read the paper as stand-alone, because many details are lacking. Still the paper is very long, with abundant figures and tables. Since this is a "data" paper, I can understand this approach. Nevertheless, it would be good to present clarifications on a number of points, or at least point the reader to the correct papers

for reference. This holds specifically for the following points:

1) The state vector. The paper mentions a state vector of emissions and concentrations. In more detail: NOx, CO emissions, lightning NOx, SO2 emissions and 35 chemical species. The latter surprises me, because the observations to not contain information on these 35 species. Probably this is outlined in an earlier paper, but this needs to be clarified/referenced to properly.

2) The weighting between emissions and concentration updates. This approach is unique in the sense that both emissions and concentration fields are in the state. However, with a assimilation window of only two hours there is a serious problem, specifically for CO (and, like the authors falsely claim, for O3). Emissions are only "seen" for two hours by the system. Still the authors present a detailed analysis of the emission increments and emission time series. But if also the concentrations in the model are adjusted, I wonder what happens to the mass balance? The concentration updates are not propagated to emissions. Likely these considerations are part of earlier papers, but need to be outlined here to some extend.

Further questions and remarks are given in the annotated pdf. These remarks also point to some obvious unit errors in figure 4. All in all, however, this paper presents the re-analysis in a fair and objective way, and points to further improvements needed. I hope my comments help to improve the paper further.

Please also note the supplement to this comment:
https://www.earth-syst-sci-data-discuss.net/essd-2020-30/essd-2020-30-RC2-supplement.pdf

———————————————————

[Figure]

**Supplement:**

[revised manuscript text omitted]

---

## Author Comment (AC1) · 25 Jul 2020

**Author's comments in reply to the anonymous referee for "An updated tropospheric chemistry reanalysis and emission estimates, TCR-2, for 2005–2018" by K. Miyazaki et al.**

We want to thank the referees for the helpful comments. We have revised the manuscript according to the comments, and hope that the revised version is now suitable for publication. Below are the referee comments in italics with our replies in normal font.

*Reply to Referee #1*

*General comments 1. Section 2.2. The covariance matrices of observation and background error are very important for the accuracy of forecasts. Please clarify how the uncertainties are determined in the text.*

The following sentences have been added to Section 2.2.
"Then background error covariance is obtained from an ensemble forecast with the updated analysis ensemble, whereas the observation error is obtained from the satellite retrieval uncertainty information (c.f., Section 3.1)."

For the NO2 retrievals, the following information has been added:
"The retrieval uncertainty of individual pixels was calculated based on error propagation in the retrieval."

*2. Section 3.1. The super-observation approach is applied to NO2 and CO, but not other species. Is there any specific reason for this?*

Super observations were not generated for the TES ozone and MLS ozone and HNO3 retrievals, because of relatively large spatial representativeness of the vertically-integrated information primarily within the free troposphere for the TES measurements and of the upper tropospheric and lower stratospheric concentrations for the MLS measurements. This is clearly explained in the revised manuscript.

The super-observation approach was also applied for the OMI $SO_2$ measurements, which is mentioned in the revised manuscript.

*3. Page 12, line 340, the authors tabulate the bias of modeled NO2 compared to OMI, SCIAMACHY, and GOME-2 observations. I notice that the consistency is significantly better for results relying on single species (e.g., DECSO in Ding et al.) What could be the driver for the bias? Is the uncertainty of the*

*assimilated system or the conflict caused by satellite observations of different species used for assimilation?*

We couldn't find any other results showing smaller biases against assimilated NO2 measurements including the DECSO paper. Nevertheless, there could be a couple of reasons for the remaining biases in general as below.

Firstly, the resolution of our global model is relatively coarse than that of other regional systems including the DECSO. Our recent study (Sekiya et al., in review) demonstrated that increasing model resolution from 1.1 degree to 0.56 degree significantly data assimilation analysis errors of tropospheric NO2 against assimilated measurements by 5–24%.

Secondly, model performance can affect NO2 analysis bias in data assimilation, as demonstrated by Miyazaki et al (2020b) using a multi-model data assimilation framework. The negative NO2 biases in the TCR-1 and TCR-2 reanalyses over the polluted regions could be associated with errors in the model chemical equilibrium states, planetary boundary layer (PBL) mixing, and diurnal variations of chemical processes and emissions. Adjusting additional model parameters such as VOC emissions, deposition, and/or chemical reactions rates by adding observational constraints could help to reduce model errors.

Thirdly, rather than degrading the NO2 analysis, multiple-species measurements provide important information for improving surface $NO_x$ source estimations and improve the chemical consistency including the relation between concentrations and the estimated emissions (Miyazaki and Eskes, 2013; Miyazaki et al., 2017). The improved representation of $NO_x$ emissions was confirmed by the better agreement of simulated ozone concentrations with independent ozonesonde observations using $NO_x$ emissions from multiple-species assimilation than those using $NO_x$ emissions from $NO_2$-only data assimilation (Miyazaki et al., 2017).

To discuss the possibilities briefly, the following sentences have been added:
"The remaining negative biases could be associated with errors in the model chemical equilibrium states, planetary boundary layer (PBL) mixing, and diurnal variations of chemical processes and emissions. Meanwhile, Sekiya et al. (2020) demonstrated that increasing model resolution from 1.1 degree to 0.56 degree reduced the analysis errors of tropospheric NO2 by 5--24%."

*Line 357, Over India, the model negative bias increased with year because of the lack of the emission increases in the a priori emissions. I suggest adding some analysis about the uncertainty associated with the a priori emissions in the text as well.*

The following sentences have been added to discuss the negative biases over India:
"The a posteriori NOx emissions in 2018 are up to 90% larger than the a priori emissions over polluted areas at grid scale, whereas the remaining negative NO2 biases suggest that the NOx emission analysis increments are insufficient. We applied a covariance inflation to the emission factors to prevent covariance underestimation caused by the application of a persistent forecast model, by inflating the spread to a minimum predefined value (i.e., 30% of the initial standard deviation (=40%)) at each analysis step. The inflation was essential to maintain emission variability and continue to increase the emissions. The remaining model biases suggest requirements for a stronger covariance inflation, although too large inflation can cause unstable analysis increments."

*Section 7.2. As mentioned by the authors VOCs have a pronounced influence on the tropospheric chemistry. Any specific reason for excluding satellite-observed VOC into the assimilated system?*

The main reason is the relatively large uncertainty in the current HCHO satellite products for decadal reanalysis (OMI and GOME-2). In most previous studies, these satellite data have been used as monthly averages to provide systematic constraints on emissions with reduced noises. Because of the short-assimilation window, it is not appropriate to use monthly mean data in our assimilation system. With the recent developments in satellite products (e.g., TROPOMI HCHO and CrIS isoprene), we expect that we will be able to update VOC emissions in the TCR-2 framework.

*Specific comments: 1. Page 3, line 58, the description of IFS and the following discussion about reanalysis pop up here. What is the relationship between TCR and those models?*

These systems are not related to the TCR-2 system, but have been compared with the TCR reanalyses and thus are useful to introduce here. To clarify this, the sentence has been rewritten as:
"Apart from the TCR systems, employing…."

*2. Page 21, line 625, the negative increments over India seems to be inconsistent with the recent report. please try to clarify the reason for such inconsistency.*

What we show here is data assimilation increment, not trend or temporal changes. Because the increments strongly depend on the a priori emissions, they can differ among data assimilation systems. To clarify this, "The negative increments" have been replaced with "The negative data assimilation increments".

*3. Fig 8 & 9, the legend is too small to see.*

Increased.

*4. Fig 11, I recommend adding the r and RMSE in the figures.*

Added.

*5.The authors performed comprehensive evaluations for multiple species using multiple measurements. It is very impressive, but easy for readers to get lost. I recommend a table to summarize the validation results by species and sources of measurements. It would be easier for readers to follow the improvement by listing the global average difference between aprior and post simulations against measurements.*

Thank you for the suggestion. Table 10 has been added to summarize the key global statistics of NO2, CO, and ozone evaluation results.

---

## Author Comment (AC2) · 25 Jul 2020

**Author's comments in reply to the anonymous referee for "An updated tropospheric chemistry reanalysis and emission estimates, TCR-2, for 2005–2018" by K. Miyazaki et al.**

We want to thank the referees for the helpful comments. We have revised the manuscript according to the comments, and hope that the revised version is now suitable for publication. Below are the referee comments in italics with our replies in normal font.

*Reply to Referee #2*

*General comments:*
*This paper presents a re-analysis of the tropospheric composition for the period 2005- 2018. This re-analysis is constructed by assimilating a wide range of satellite observations in a high-resolution chemistry-transport model. This is a great effort that should find its spin-off in the community. The paper is based on a large number of underlying studies that outline the method in more detail. This makes it difficult to read the paper as stand-alone, because many details are lacking. Still the paper is very long, with abundant figures and tables. Since this is a "data" paper, I can understand this approach. Nevertheless, it would be good to present clarifications on a number of points, or at least point the reader to the correct papers for reference.*

According to the comments, more detailed information has been added, as described below.

*This holds specifically for the following points:*
*1) The state vector. The paper mentions a state vector of emissions and concentrations. In more detail: NOx, CO emissions, lightning NOx, SO2 emissions and 35 chemical species. The latter surprises me, because the observations to not contain information on these 35 species. Probably this is outlined in an earlier paper, but this needs to be clarified/referenced to properly.*

To more clearly describe the state and control vectors used in this study, the following sentences have been added:
"The control vector, $\mathbf{z} = \mathbf{D}\,\mathbf{x}$, is a subset of the state vector $\mathbf{x}$ to be adjusted during assimilation, where $\mathbf{D}$ is a mapping matrix. The control vector $\mathbf{z}$ is updated at every analysis step by observations and then mapped back to the state vector $\mathbf{x}$. Some variables in the state vector $\mathbf{x}$ are not parts of the control vector $\mathbf{z}$ for theoretical and practical reasons as discussed below in this section."
"The optimization of the variable localization was based on a comparison against independent satellite and aircraft data, as described in Miyazaki et al., (2015). The analysis increments through the NO2 assimilation

were limited to adjusting only the surface emissions of NOx, LNOx sources, and concentrations of NOy species (=NOx+HNO3+HNO4+PAN+MPAN+N2O5). The MOPPIT CO and OMI SO2 measurements were used for constraining surface CO and SO2 emissions only, respectively. For the LNOx sources, covariances with MOPITT CO data were neglected. Concentrations of NOy species and ozone were optimized from TES ozone, OMI, SCIAMACHY, and GOME-2 NO2, and MLS ozone and HNO3 observations. Although the concentrations of VOCs are included in the state vector, they were not included in the control vector and thus were not optimized in the current setting, because the current assimilated data sets did not improve their fields obviously. Consequently, the control vector in this study includes the concentration of NOx, HNO3, HNO4, PAN, MPAN, N2O5, and ozone, as well as the NOx, SO2, and CO emission sources. "

As mentioned here, although the chemical concentrations of 35 species were included in the state vector, some of them were not included in the control vector and thus were not optimized through data assimilation in the current setting. The relevant sentences have been rewritten as follows in the revised manuscript:

The main text:
"The state vector includes several emission sources (surface emissions of NOx, CO and SO2, and lightning NOx (LNOx) sources) as well as the concentrations of 35 chemical species (c.f., Fig. 3 in Miyazaki et al., 2012). As described above, limited variables were included in the control vector and optimized by applying covariance localization in the reanalysis calculations."

*2) The weighting between emissions and concentration updates. This approach is unique in the sense that both emissions and concentration fields are in the state. However, with a assimilation window of only two hours there is a serious problem, specifically for CO (and, like the authors falsely claim, for O3). Emissions are only "seen" for two hours by the system. Still the authors present a detailed analysis of the emission increments and emission time series. But if also the concentrations in the model are adjusted, I wonder what happens to the mass balance? The concentration updates are not propagated to emissions. Likely these considerations are part of earlier papers, but need to be outlined here to some extend.*

In the TCR-2 calculation, only CO emissions were optimized using MOPITT CO, whereas the CO concentrations were not adjusted by the data assimilation analysis, which is clearly described in the revised manuscript. Our validation results against the aircraft measurements confirm that the constraints provided for the surface emissions are propagated well into the concentrations of the entire troposphere, while keeping the mass balance in the atmosphere (except for chemical losses and productions of CO). Even in the short assimilation window, data assimilation increments can be used to measure systematic model

biases in emissions that could affect long-term model errors. Same for SO2 emissions. For NOx, concentration adjustments are quickly lost in the lower atmosphere due to the short lifetime, while emission adjustments are more efficient to store the information over longer time periods. The following sentences have been added:

"Even in the short assimilation window (i.e., two-hour), data assimilation increments can be used to measure systematic model biases in emissions that could affect long-term model errors."

"For NOx, concentration adjustments are quickly lost in the lower atmosphere due to the short lifetime, while emission adjustments are more efficient to store the information over longer time periods (Miyazaki and Eskes, 2013; Miyazaki et al., 2017)."

*Further questions and remarks are given in the annotated pdf. These remarks also point to some obvious unit errors in figure 4. All in all, however, this paper presents the re-analysis in a fair and objective way, and points to further improvements needed. I hope my comments help to improve the paper further.*

Thank you for the helpful comments. We have endeavored to address your helpful concerns and comments in the revised manuscript. Please see additional replies to your comments below.

*L9 This sentence is strange. "Played" sounds like it stopped doing this...Please make clear what you intend to say here.*

Replaced by "which was important in improving…."

*L10 I think it is not the products, but the system that helps this understanding.*

Replaced by "chemical reanalysis".

*L99 I guess it might follow later, but what about natural emissions?*

Natural emissions include biomass burning for CO and soil and lightning emissions for NOx, which are already mentioned in the manuscript. Please also see the reply below (L117).

*L116 Which ones*

The sentence has been rewritten as "35 chemical species (c.f., Fig. 3 in Miyazaki et al. (2012b))". Please also see our reply above.

*L117 So, no natural emissions in the state?*

Natural sources include biomass burning for CO and soil and lightning emissions for NOx, which are included in the state vector, as described in the manuscript. Natural CO sources also include chemical productions from the oxidation of hydrocarbons by the incomplete combustion of fossil fuels and biofuels, which is not included in the state vector. The following sentence has been added:
"The emissions include both anthropogenic and natural (i.e., soil and biomass burning) sources, except for chemical productions of CO by the oxidation of methane and biogenic non-methane hydrocarbons (NMHCs)."

*L152 here I wonder why then not to TES O3?*

The following sentence has been added:
"Super observations were not generated for the TES retrievals, because of relatively large spatial representativeness of the vertically-integrated information primarily within the free troposphere."

*L161 Probably stated in other papers, but good to mention what is done for CH4...*

The following sentence has been added:
"Methane concentrations were scaled on the basis of present-day values with reference to the surface concentration."

*L162 Here I wonder why SO2 emissions are not optimized (apparently only the atmospheric state is optimized). Given the nature of the gas (e.g. volcanic eruptions) this would make sense?*

SO2 emissions are optimized, which is explained in the revised manuscript.

*L252 removed "the most model negative"*

Replaced by "removed most of the model biases"

*L256 But this could also be due to changes in the chemistry, due to updated (better) CO & NOx?*

At high latitudes, the influences of changes in chemistry is considered to be limited because of inactive photochemistry and lack of direct observational constraints on CO and NOx, whereas the relatively long chemical lifetime of ozone would explain the large changes via atmospheric transports.

*L298 well, this reasoning I never understand: it shows that it has similar information, so the added value will not be large?? Or do you mean that is can replace some of the existing assimilated observations?*

The evaluation of new satellite products, that are not assimilated, using the existing data assimilation system can be used to provide long-term global and regional statistics of the new satellite data in "monitoring phase" without any additional cost in data assimilation calculation as presented here. If the monitoring shows reasonable data quality (e.g., no rapid changes in departures), the data could be used in following "data assimilation phase" more safely in data assimilation. This monitoring to assimilation step is common in NWP and other data assimilation developments.

*L383 But I understand that you both update emissions AND concentrations. This remains vague in the paper. How much is ascribed to emissions and how much to concentration adjustments?*

For short-lived species such as NOx discussed here, the analysis increment made to the concentrations can partly quickly be lost after the forecast, and they do not affect the emission estimates substantially, as discussed in Miyazaki and Eskes (2013) and Miyazaki et al. (2017). The proportion of emission adjustments to concentration adjustments at each level depend on vertical sensitivity of assimilated measurements and background errors for emissions and concentrations as estimated from ensemble simulations at each grid point. For CO, please see my reply above.

*L418 Was not clear that SO2 emissions were in the state. I thought only CO and NOx?*

SO2 emission were included in the state vector. This is clearly mentioned in the revised manuscript.

*L425 Check units in figure!*

Thank you for checking the units. Corrected.

*L439 ????*

Replaced by

"we focus on regions over and downstream of highly polluted areas only"

*L466 so VOC emissions are not in the state, but what about the possibility to adjust the 3D fields....?*
*You mention on line 117 that 35 species' concentrations are in the state. How can these all be*
*constrained without suitable observations....as a stand-alone paper, there remain many unclear issues...*

To clarify this point, the following sentence has been added:

"Although the concentrations of VOCs are included in the state vector, they were not included in the control vector and thus were not optimized in the current setting, because the current assimilated data sets did not improve their fields obviously."

Please also see my reply above.

*L485 during what....and unit seems incorrect.*

Replaced by "during the reanalysis periods". Corrected the unit.

*L489 In that sense it would be good if you add some information about CH4 i your system.*

Although the current setting uses a fixed methane fields (please see our reply above), our chemical reanalysis approach has the potential to improve methane simulations/inversion in an interactive system.

*L528 AOD is an aerosol observation.*

Replaced by "AOD and aerosol concentration observations".

*L548 2x???*

The model includes the heterogeneous $HO_2$ loss by aerosols and cloud droplets. However, it assumes the final product of this $HO_2$ reaction to be $H_2O_2$ not $H_2O$. The absence of this loss process could lead to the overestimations in $H_2O_2$ and $HO_2$, as suggested by Mao et al. (2013). In addition, errors in the removal of $HO_2$ by wet deposition processes might also cause biased concentrations.

*L553 droplets.*

Corrected.

*L556 CH2O OMI could be used????*

This is discussed in Section 7.

*L571 This is only true of you can disentangle the contribution of emissions from concentration adjustments.*

Yes. Please see our reply above.

*L579 I assume that no spatial and temporal correlations are used in the prior emissions (this is common in most emission estimate applications)....*

Yes.

*L605 Tg CO/year*

Corrected.

*L655 Anyhow, the lack of emissions over oceans needs to be discussed....(I am not an expert, but knwo that lightning happens over oceans also).*

There are lightning sources over oceans too, as shown in the figure, but they are much weaker than over lands.

*L719 This sounds weird....and needs reqriting.*

What we meant here is that the number of available ozonesonde data after 2017 was limited at the time of this research, not actual changes in the ozonesonde network. The evaluated results can strongly dependent on changes in the number and location of observations used for validation, because the representativeness of each individual profiles for regional mean ozone concentrations is limited, as demonstrated in our previous study (Miyazaki and Bowman, 2017). To clearly explain them, the sentence has been rewritten as:

"The mean ozonesonde concentrations at SH mid latitudes show rapid changes after 2017, which were associated with the reduced number of available ozonesonde observations at the time of this research and consequent increased representativeness errors of the ozonesonde network for the large domain."

*L727 I would add also the limited representation of atmospheric chemistry....*

Added.

*L740 In that sense it would be good to have a list of tracer of which the atmospheric abundance is adjusted....are this 35?*

The information is more clearly provided in the revised manuscript. Please also see our reply above.

*L765 what about the role of MOPITT in covering high latitudes? Is there a problem in winter? What coverage did you include?*

The following sentences have been added:
"We exclude MOPITT data in polar regions (>65°), where the quality deteriorates and the information content lowers. because of potential problems related to cloud detection and icy surfaces. We also excluded the night-time data using a filter based on solar zenith angle, because daytime conditions typically provide better thermal contrast conditions for the retrievals."

*L773 well, the lifetime of ozone is considered to be longer than that of CO, but this calculation omits the rapid cycling between NO + O3 <--> NO2 + O3 (the pohotostationary state).   so, this is tricky ...NOx OK....but ozone is also not emitted, so the adjustments in the state need some horizontal and vertical weighting functions....which is surely described in other papers.*

The lifetime of ozone (or Ox) is considered to be (much) shorter than that of CO in the lower troposphere for most regions. The lifetime of ozone is approximately 23 days (Young et al., ACP, 2013) for the tropospheric average with much shorter lifetime (- a few days) in the lower troposphere, whereas that of CO is a few weeks to 2 months (Holloway et al., JGR, 2000). Rapid ozone changes due to fast photochemical reactions in the lower troposphere are considered to be localized well in both time and space, which can be optimized using local measurements. Meanwhile, we expect that the photochemical equilibrium state are also adjusted thought the comprehensive constraints on the tropospheric chemistry system including OH. For situations with relatively long chemical lifetime of ozone such as in the middle

and upper troposphere, data assimilation analysis can be sensitive to horizontal localization length. The horizontal localization length has been optimized from sensitivity calculations, where the vertical localization is based on the retrieval averaging kernel. The following sentences have been added. "The cut-off radius was set to 1,643 km for NOx emissions and 2,019 km for CO emissions, lightning sources, and chemical concentrations based on sensitivity calculations. However, the optimal localization length may depend on the location, season, and species, reflecting meteorological conditions and the chemical lifetime."

*L775 again: what is done with methane in the system is relevant here.*

Please see our reply above.

*L826 sound too optimistic given the large problems at high latitude NH....*

As summarized in Table 5 and stated here, this is true for the SH, the tropics, and NH mid latitudes. NH high latitudes are not included in this statement.

*Fig 2, 4,17*

Corrected.